# Neuro-Fuzzy Concept Learning for Interpretable Large Multimodal Models

**Ritik Mishra** [1]  **Vanshika Gupta** [* 2]  **M. Sajid** [* 1]  **M. Tanveer** [1]

## Abstract

Large Multimodal Models (LMMs) integrate uni-modal encoders with Large Language Models (LLMs) to execute complex multimodal tasks. Despite progress in the field, understanding the internal representations of these models through interpretable logic remains an open problem. To address this, we present a framework utilizing a Human-Inspired (Neuro-fuzzy) approach for learning token representations. In this method, we leverage fuzzy rules to compute activation firing strengths, which are subsequently defuzzified to extract distinct concepts. This mechanism allows for the interpretation of learned representations directly through explicit logic. Consequently, we derive "multimodal concepts" that are both semantically coherent and interpretable. We validate our approach through rigorous qualitative and quantitative experiments, demonstrating the utility of these concepts in interpreting test samples. Additionally, we evaluate the disentanglement of the learned concepts and the efficacy of their grounding in both visual and textual domains. The source code is available at https://github.com/mtanveer1/Neuro-FeX-LMM.

## 1. Introduction

Large Multimodal Models (LMMs) have achieved remarkable success by integrating visual encoders with large language models to jointly reason over images and text (Alayrac et al., 2022; Laurençon et al., 2023). These models excel at complex tasks such as visual question answering, image captioning, and visual reasoning. However, their internal decision-making processes remain opaque, rais-

ing critical questions about reliability and trustworthiness (Jacovi et al., 2021; Markus et al., 2021). As LMMs are increasingly adopted in high-stakes applications, the need for interpretable explanations of their predictions has become paramount.

Most existing explainability approaches for deep networks rely on attribution methods, which assign relevance scores to input features through gradient-based or perturbation-based techniques (Smilkov et al., 2017; Fel et al., 2023b). While these methods effectively highlight where a model attends in the input space, they fail to convey what semantic concepts drive the prediction (Sixt et al., 2020; Colin et al., 2022). Attribution maps identify spatially important regions but abstract away the underlying semantic structure, leaving stakeholders unable to understand which high-level concepts the model relies on for its decisions (Slack et al., 2021).

Concept-based explainability offers a promising alternative by decomposing neural representations into interpretable, high-level semantic factors (Kim et al., 2018). Methods such as Concept Activation Vectors (CAVs) (Kim et al., 2018) and Concept Bottleneck Models (Koh et al., 2020) extract human-aligned concepts from learned representations, while recent dictionary learning approaches apply mathematical decomposition techniques such as Non-negative Matrix Factorization (NMF), Principal Component Analysis (PCA), and Semi-Non-negative Matrix Factorization (Semi-NMF) to factorize neural activations (Parekh et al., 2024; Fel et al., 2023a;c; Gandelsman et al., 2024; Bhalla et al., 2024). However, these approaches face fundamental limitations: they typically rely on purely algebraic factorizations that decompose activations into basis vectors or linear directions (Ghorbani et al., 2019; Yeh et al., 2020), and critically, they lack an explicit reasoning structure that connects concepts to predictions in a cognitively interpretable manner (Colin et al., 2022; Miller, 2019). Whether requiring extensive concept annotations (Koh et al., 2020; Margeloiu et al., 2021) or operating through unsupervised decomposition (Ghorbani et al., 2019; Fel et al., 2023c), existing methods treat concept extraction as a purely mathematical problem, divorced from the logical reasoning processes and rule-based structures that humans naturally employ when making decisions (Lipton, 2018; Rudin, 2019). Decomposing representations into interpretable components is valuable, but without exposing

---

*Equal contribution. [1]Department of Mathematics, Indian Institute of Technology Indore, Madhya Pradesh, 453552, India [2]Department of CSE, Indian Institute of Technology Indore, Madhya Pradesh, 453552, India. Correspondence to: Ritik Mishra <phd2301241003@iiti.ac.in, ritik.aimath@gmail.com>.

*Proceedings of the 43$^{rd}$ International Conference on Machine Learning*, Seoul, South Korea. PMLR 306, 2026. Copyright 2026 by the author(s).

how these concepts interact through explicit logical rules, such methods provide limited insight into the model's actual decision-making process (Jacovi et al., 2021; Doshi-Velez & Kim, 2017).

We propose bridging this gap through a brain-inspired neuro-fuzzy (Jang & Sun, 1995) framework that unifies the learning capabilities of LMMs with the transparent reasoning structure of fuzzy logic systems. Unlike purely algebraic approaches, neuro-fuzzy systems express decisions through interpretable "IF–THEN" rules with explicit antecedents and consequents (de Campos Souza, 2020; Sajid et al., 2024; 2025). This dual nature: neural adaptability with logical transparency, enables a fundamentally different approach to interpretability: rather than simply identifying which concepts are present, we can inject the capability of how concepts combine through fuzzy inference rules to produce model predictions. By treating LMM representations as outputs of an implicit fuzzy inference system, we transform concept learning from a dimensionality-reduction problem into a rule-extraction problem that mirrors human cognitive processes.

**Conflict of Interest Disclosure.** The authors declare no financial conflicts of interest related to this work.

## 2. Motivation and Contributions

Understanding why an LMM produces a particular prediction requires more than identifying important concepts; it requires understanding the logical structure governing how concepts interact and contribute to outputs (Grobrügge et al., 2025). Thus, the gap between concept identification and reasoning structure limits the practical utility of existing explainability methods.

Neuro-fuzzy systems, particularly those based on Takagi-Sugeno-Kang (TSK) inference, provide a principled framework for encoding such reasoning. In TSK systems, knowledge is represented through fuzzy IF–THEN rules where antecedents capture input conditions through fuzzy membership functions and consequents define local linear models. Each rule captures a specific reasoning pattern, and the final output emerges from aggregating rule contributions weighted by their activation strengths, a process called defuzzification. This architecture is inherently interpretable: stakeholders can trace which rules activated, with what strength, and how they collectively determined the prediction. We hypothesize that LMM latent representations encode an implicit neuro-fuzzy structure. Specifically, we posit that feature embeddings can be decomposed into a linear combination of neuro-fuzzy concepts, where each concept corresponds to the consequent parameter of a latent fuzzy rule. The activation patterns in the network then reflect the normalized firing strengths of these rules.

Thus, we propose an interpretable LLM, Neuro-FeX, a unified neuro-fuzzy concept-learning framework that interprets large multimodal models via brain-inspired fuzzy inference structures. Our approach differs fundamentally from existing concept extraction methods in three key ways. First, we impose a hybrid neuro-fuzzy architecture that combines two complementary decomposition stages: fuzzification (mapping representations to fuzzy rule activations) and defuzzification (reconstructing outputs from rule consequents). Second, we extract multimodally grounded concepts that are interpretable in both visual and linguistic modalities, enabling richer semantic understanding. Third, our framework provides rule-level interpretability, exposing not just which concepts are present but how they interact through fuzzy inference to produce predictions.

Our key contributions are:

- We propose the first unified neuro-fuzzy interpretability framework for large multimodal models, **Neuro-FeX**, which decomposes latent representations via a two-stage fuzzification–defuzzification process, providing both rule activation transparency and concept-level interpretability.

- We develop an optimization framework that integrates Semi-NMF dictionary learning with fuzzy inference constraints, extracting concepts as consequent parameters of a TSK fuzzy system with normalized rule firing strengths and linear polynomial consequents.

- We show that the learned neuro-fuzzy concepts exhibit meaningful multimodal grounding across diverse semantic categories, supported by qualitative visualizations and quantitative evaluations.

## 3. Approach

### 3.1. Large Multimodal Model Architecture

**Model Components.** Our framework applies to large multimodal models $f$ built from three modules: a frozen or pretrained visual encoder $f_V$ that processes images, a connector network $C$ that bridges modalities, and a large language model $f_{LM}$ with $N_L$ transformer layers. We work with models trained on image-to-text generation using a dataset $S = \{(X_i, y_i)\}_{i=1}^N$ containing image-caption pairs, where $X_i \in \mathcal{X}$ represents the image space and $y_i \subset \mathcal{Y}$ represents token sequences from the vocabulary.

**Forward Pass and Token Representations.** Given an input image $X$, the visual encoder produces features that the connector transforms into $N_V$ visual token embeddings. These visual tokens, concatenated with text token embeddings, form the input sequence $h_1, h_2, \ldots, h_p$ to $f_{LM}$. We denote the hidden state at position $p$ in layer $l$ as $h_p^{(l)}$, which ac-

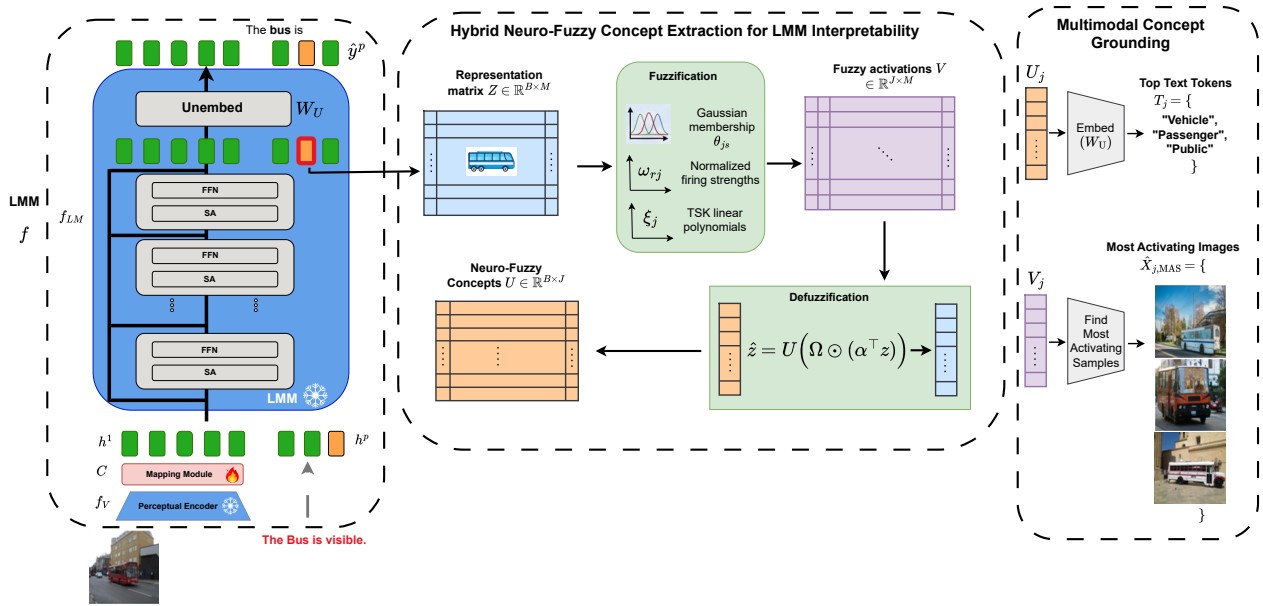

*Figure 1.* **Overview of Hybrid Neuro-fuzzy concept extraction and grounding in Neuro-FeX.** Given a pretrained captioning LMM and a target token (e.g., *bus*), our method extracts token-specific internal representations from the model across a collection of images. These representations are aggregated into a matrix $Z$, which is then decomposed into neuro-fuzzy concepts $U$ and corresponding activations $V$. Each concept $u_j \in U$ is grounded in both textual and visual modalities. Textual grounding is obtained by decoding $u_j$ through the unembedding matrix $W_U$ to produce a set of high-probability words $T_j$, while visual grounding is derived from the activations $v_j$ by identifying the maximally activating samples $X_{j,\text{MAS}}$.

cumulates information through the transformer's residual connections. At the input layer, $h_p^{(0)} = h_p$.

For autoregressive text generation, positions beyond the visual prefix ($p > N_V$) predict the next token as:

$$\hat{y}_p = f_{LM}(h_1, h_2, \ldots, h_{N_V}, \ldots, h_p), \quad (1)$$

where the visual prefix is computed as $h_1, \ldots, h_{N_V} = C(f_V(X))$, and subsequent text tokens are embedded as $h_p = \text{Emb}(\hat{y}_{p-1})$ for $p > N_V$. Generation starts with a special start token at position $N_V + 1$ and continues until an end token is predicted. The final prediction $\hat{y}_p$ results from projecting the top-layer representation $h_p^{(N_L)}$ through an unembedding matrix $W_U$ followed by softmax normalization.

**Training Paradigm.** LMMs are typically trained with cross-entropy loss on next-token prediction, learning to generate captions conditioned on visual input. Many architectures freeze both $f_V$ and $f_{LM}$ while training only the lightweight connector $C$, a setup that exhibits generalization despite minimal parameter updates. While some recent models fine-tune $f_{LM}$ for improved performance, our method applies to both configurations. We primarily analyze frozen-LLM architectures as they provide clearer insights into how pre-trained language models adapt to multimodal reasoning.

**Residual Stream Interpretation.** Following the interpretability framework of (Elhage et al., 2021), we view

transformer layers as performing additive updates to a central representation stream. At each layer, the representation $h_p^{(l)}$ receives contributions from attention and feedforward components:

$$h_p^{(l+1)} = h_p^{(l)} + a_p^{(l)} + m_p^{(l)}, \quad (2)$$

where $a_p^{(l)}$ represents the attention block's contribution, computed causally over previous positions, and $m_p^{(l)}$ represents the MLP block's contribution. The MLP applies two linear layers with a nonlinearity $\sigma$ between them, processing the combined input $h_p^{(l)} + a_p^{(l)}$. This additive structure allows us to trace information flow and decompose representations into interpretable components.

### 3.2. TSK Fuzzy Inference System

The TSK fuzzy system provides a principled framework for modeling complex nonlinear systems through interpretable fuzzy rules. Unlike purely black-box approaches, TSK systems combine fuzzy logic reasoning with local linear models, enabling both better approximation and human-interpretable decision structures. This dual capability makes TSK a powerful tool for applications requiring transparency and accountability.

We denote a collection of latent representations as $\{\mathbf{z}_r\}_{r=1}^M$, where each $\mathbf{z}_r \in \mathbb{R}^{B \times 1}$ represents a feature vector extracted

from the $r$-th sample. Here, $B$ denotes the dimensionality of the representation space, and $M$ represents the total number of samples in the dataset. We organize these representations into a matrix $\mathbf{Z} = [\mathbf{z}_1, \mathbf{z}_2, \ldots, \mathbf{z}_M] \in \mathbb{R}^{B \times M}$, where columns correspond to individual samples and rows correspond to features.

A TSK system is defined by a set of $J$ fuzzy IF-THEN rules. The $j$-th rule ($j \in \{1, \ldots, J\}$) takes the form:

$$\text{Rule } j: \quad \text{IF } z_{r,1} \text{ is } F_{j1} \text{ AND } \ldots \text{ AND } z_{r,B} \text{ is } F_{jB}$$
$$\text{THEN } v_{rj} = \xi_j(z_{r,1}, \ldots, z_{r,B}), \quad (3)$$

where $z_{r,s}$ denotes the $s$-th feature of input sample $\mathbf{z}_r$ (for $s \in \{1, \ldots, B\}$), $F_{js}$ represents a fuzzy set characterizing the $s$-th feature in the $j$-th rule, and $\xi_j : \mathbb{R}^B \to \mathbb{R}$ is a consequent function, typically chosen as a linear or polynomial function of the inputs.

**Fuzzy Inference Process.** For a given input $\mathbf{z}_r$, the inference process consists of three stages:

*(1) Fuzzification:* Each feature $z_{r,s}$ is evaluated against $F_{js}$ using a membership function $\Theta_{js} : \mathbb{R} \to [0, 1]$, which quantifies the degree to which $z_{r,s}$ belongs to $F_{js}$.

*(2) Rule Activation:* The firing strength (or activation level) of the $j$-th rule for input $\mathbf{z}_r$ is computed by aggregating membership degrees across all features: $T_{rj} = \prod_{s=1}^{B} \Theta_{js}(z_{r,s})$. This product operation implements the fuzzy AND connective, ensuring that a rule activates only when all antecedent conditions are simultaneously satisfied to some degree.

*(3) Defuzzification:* The final output $\hat{\mathbf{z}}_r$ is obtained by computing a weighted average of rule consequents, where weights correspond to normalized firing strengths: $\hat{\mathbf{z}}_r = \frac{\sum_{j=1}^{J} T_{rj} \cdot v_{rj}}{\sum_{j=1}^{J} T_{rj}} = \sum_{j=1}^{J} \alpha_{rj} \cdot v_{rj}$, where $\alpha_{rj} = \frac{T_{rj}}{\sum_{j=1}^{J} T_{rj}}$ represents the normalized firing strength of rule $j$ for sample $r$. This formulation ensures that rules with higher activation contribute more strongly to the final output.

# 4. Proposed Work

## 4.1. Method Overview

Given a pretrained large multimodal model $f$ and a target token of interest $t \in \mathcal{Y}$, our Neuro-FeX framework extracts interpretable neuro-fuzzy concepts through a three-stage:

1. **Representation Extraction:** We identify images from dataset $S$ where token $t$ appears in both the ground-truth caption and model prediction. For each such image, we extract the latent representation $h_p^{(L)}$ at the position where $t$ is first predicted, constructing a representation matrix $\mathbf{Z} \in \mathbb{R}^{B \times M}$ containing $M$ sample representations of dimension $B$.

2. **Hybrid Neuro-Fuzzy Decomposition Concept Learning:** We decompose $\mathbf{Z}$ using a two-component architecture inspired by TSK fuzzy inference:

   - *Component 1 - Fuzzification:* Projects the representation matrix $\mathbf{Z}$ into a fuzzy activation space, computing normalized firing strengths $\mathbf{V} \in \mathbb{R}^{J \times M}$ for $J$ latent fuzzy rules across $M$ samples.

   - *Component 2 - Defuzzification:* Reconstructs $\mathbf{Z}$ from fuzzy activations by extracting a neuro-fuzzy concept dictionary $\mathbf{U} \in \mathbb{R}^{B \times J}$, where each column represents the consequent parameter of a fuzzy IF-THEN rule.

3. **Multimodal Concept Grounding:** We establish semantic interpretability for extracted neuro-fuzzy concepts by grounding them in both visual and textual modalities, comprehensible explanations.

Our objective is to understand the internal reasoning structure of LMMs rather than explain final predictions. The extracted neuro-fuzzy concepts reveal the implicit rule-based logic encoded within the model's latent representations.

## 4.2. Representation Extraction from Target Tokens

To extract semantically meaningful representations for a target token $t$, we construct a filtered dataset $\mathcal{X}$ containing only images where $t$ appears in both the model's prediction $\hat{y}$ and the ground-truth caption $y_i$. This dual-occurrence criterion ensures that our extracted representations (1) reflect the model's internal processing of $t$, and (2) maintain alignment with true semantic content. Formally:

$$\mathcal{X} = \{X_i \mid t \in f(X_i) \cap y_i, \text{ for } (X_i, y_i) \in S\}. \quad (4)$$

For each image $X \in \mathcal{X}$, we identify the first position $p > N_V$ in the predicted sequence where token $t$ is generated, i.e., where $\hat{y}_p = t$. At this position, we extract the deep-layer representation $h_p^{(L)}$ from layer $L$ of the language model. Crucially, $h_p^{(L)}$ is not merely the output of layer $L$, it represents the accumulated residual stream containing information propagated through all previous layers. Due to causal attention, this representation naturally integrates multimodal information from visual tokens $\{h_1^{(l)}, \ldots, h_{N_V}^{(l)}\}$ at earlier layers $l < L$, making it suitable for multimodal concept extraction. We stack the extracted representations as columns to form the representation matrix:

$$\mathbf{Z} = [\mathbf{z}_1, \mathbf{z}_2, \ldots, \mathbf{z}_M] \in \mathbb{R}^{B \times M}, \quad (5)$$

where $\mathbf{z}_j = h_p^{(L)}$ for the $j$-th sample in $\mathcal{X}$, $B$ denotes the hidden dimension of the language model, and $M = |\mathcal{X}|$ is the number of valid samples. This matrix serves as the input to our hybrid neuro-fuzzy decomposition algorithm.

## 4.3. Hybrid Neuro-Fuzzy Decomposition for Interpretable Reasoning

We decompose the representation matrix $\mathbf{Z}$ using a hybrid neuro-fuzzy architecture that models LMM representations as outputs of a TSK fuzzy inference system. This approach provides three interpretability advantages over algebraic methods: (1) *rule-level transparency* through $J$ distinct fuzzy rules with interpretable antecedents and consequents, (2) *activation-level understanding* via normalized firing strengths showing which rules activate for each sample, and (3) *concept-level grounding* where consequent parameters form an interpretable concept dictionary. We adopt first-order TSK consequents: $v_{rj} = \xi_j(\mathbf{z}_r) = \sum_{s=1}^{B} \alpha_{js} z_{r,s}$, where $\alpha_{js}$ are learnable coefficients. The linear form allows tracing exactly how each feature dimension contributes to rule $j$'s output, making the computational logic transparent. For antecedents, we use Gaussian membership functions:$\Theta_{js}(z_{r,s}) = \exp\left(-\left(\frac{z_{r,s}-c_{js}}{\sigma_{js}}\right)^2\right)$, where $c_{js}$ and $\sigma_{js}$ define the center and width, initialized via K-means or top singular vectors. Each center $c_{js}$ represents a prototype feature value, while width $\sigma_{js}$ controls how strictly the rule requires features to match this prototype, creating interpretable activation conditions. The normalized firing strength is: $\omega_{rj} = \frac{\prod_{s=1}^{B} \Theta_{js}(z_{r,s})}{\sum_{k=1}^{J} \prod_{s=1}^{B} \Theta_{ks}(z_{r,s})}$. This quantifies rule $j$'s relevance to sample $\mathbf{z}_r$ on a normalized scale, providing explicit evidence of which logical rules govern each representation.

### 4.3.1. COMPONENT 1: FUZZIFICATION

Fuzzification transforms neural activations into interpretable rule-firing patterns that reveal which fuzzy rules are active and their activation strengths. For sample $\mathbf{z}_r$, fuzzification computes: $\mathbf{v}_r = [\omega_{r1}v_{r1}, \omega_{r2}v_{r2}, \ldots, \omega_{rJ}v_{rJ}]^\top \in \mathbb{R}^J$. Each element represents the weighted contribution of rule $j$, enabling us to identify which semantic reasoning patterns activate for each sample rather than merely which neurons fire. The full fuzzy activation matrix is:$\mathbf{V} = [\mathbf{v}_1, \mathbf{v}_2, \ldots, \mathbf{v}_M] \in \mathbb{R}^{J \times M}$ Matrix $\mathbf{V}$ enables identifying which rules consistently co-activate, which are context-specific, and how activation patterns vary across samples.

### 4.3.2. COMPONENT 2: DEFUZZIFICATION

Defuzzification reconstructs $\mathbf{Z}$ from fuzzy activations $\mathbf{V}$ by learning a neuro-fuzzy concept dictionary $\mathbf{U}$. The defuzzified output is: $\hat{\mathbf{z}}_r = \mathbf{U}\mathbf{v}_r = \sum_{j=1}^{J} \omega_{rj} v_{rj} \mathbf{u}_j$, where $\mathbf{U} = [\mathbf{u}_1, \ldots, \mathbf{u}_J] \in \mathbb{R}^{B \times J}$ is the concept dictionary and each $\mathbf{u}_j \in \mathbb{R}^B$ is a semantic concept. This expresses the representation as a weighted combination of interpretable concepts, where weights directly reflect each rule's contri-

bution strength. For all samples:

$$\hat{\mathbf{Z}} = \mathbf{U}\mathbf{V} = \mathbf{U}\left(\mathbf{\Omega} \odot (\alpha^\top \mathbf{Z})\right) \in \mathbb{R}^{B \times M}, \qquad (6)$$

where $\mathbf{\Omega} \in \mathbb{R}^{J \times M}$ contains normalized firing strengths and $\odot$ denotes Hadamard product (element-wise multiplication). The factorization $\hat{\mathbf{Z}} = \mathbf{U}\mathbf{V}$ decomposes opaque neural activations into two interpretable components: concept dictionary $\mathbf{U}$ (what semantic content exists) and rule activations $\mathbf{V}$ (which rules fire for each sample).

The decomposition provides three key advantages: (1) each concept $\mathbf{u}_j$ can be grounded in visual and textual modalities, (2) activations $\mathbf{v}_r$ enable local interpretation by identifying which rules fired for any sample, and (3) unlike NMF or PCA, the structure explicitly models IF-THEN reasoning where high $\omega_{rj}$ means concept $\mathbf{u}_j$ contributes to the representation.

## 4.4. Optimization Framework for Neuro-Fuzzy Concept Learning

The decomposition $\mathbf{Z} \approx \mathbf{U}\mathbf{V}$ factorizes the representation matrix into two low-rank matrices: $\mathbf{U} \in \mathbb{R}^{B \times J}$ containing neuro-fuzzy concepts (rule consequents) and $\mathbf{V} \in \mathbb{R}^{J \times M}$ containing rule activations (firing strengths), where $J \ll \min(B, M)$. Each column $\mathbf{u}_j$ of $\mathbf{U}$ represents the semantic concept associated with fuzzy rule $j$, while each column $\mathbf{v}_r$ of $\mathbf{V}$ indicates which rules activate for sample $r$ and their contribution strengths.

While concept decomposition has been studied using PCA, K-Means, NMF, and Semi-NMF (Fel et al., 2023a; Parekh et al., 2024), these purely algebraic approaches lack the explicit logical reasoning structure that neuro-fuzzy systems provide. Our framework differs fundamentally by grounding the decomposition in TSK fuzzy inference, where $\mathbf{U}$ and $\mathbf{V}$ are not arbitrary basis vectors and coefficients, but rather interpretable IF-THEN rule components with clear semantic meaning.

We formulate the optimization problem as:

$$(\mathbf{V}^*, \mathbf{U}^*) = \arg\min_{\mathbf{V}, \mathbf{U}} \|\mathbf{Z} - \mathbf{U}\mathbf{V}\|_F^2 + \lambda\|\mathbf{V}\|_1 + \epsilon\|\mathbf{U}\|_F^2,$$
$$\text{subject to } \mathbf{V} \geq 0, \qquad (7)$$

where the non-negativity constraint on $\mathbf{V}$ ensures that rule activations remain interpretable as firing strengths (analogous to clustering membership (Ding et al., 2008)). The $\ell_1$ penalty on $\mathbf{V}$ enforces sparsity, reflecting the assumption that each sample activates only a subset of fuzzy rules.

**Alternating Optimization.** We solve Eq. (7) through alternating minimization, iteratively updating $\mathbf{U}$ and $\mathbf{V}$ as: *Step 1: Update Concept Dictionary $\mathbf{U}$.* Given fixed activations $\mathbf{V}$, we solve a Ridge Regression problem:

$$\mathbf{U}^{(t+1)} = \arg\min_{\mathbf{U}} \|\mathbf{Z} - \mathbf{U}\mathbf{V}^{(t)}\|_F^2 + \epsilon\|\mathbf{U}\|_F^2, \quad (8)$$

which has closed-form solution:

$$\mathbf{U}^{(t+1)} = \mathbf{Z}(\mathbf{V}^{(t)})^\top \left( \mathbf{V}^{(t)}(\mathbf{V}^{(t)})^\top + \epsilon\mathbf{I} \right)^{-1}. \quad (9)$$

This step learns neuro-fuzzy concepts that optimally reconstruct representations given current rule activations.

*Step 2: Update Rule Activations* $\mathbf{V}$. Given fixed concepts $\mathbf{U}^{(t+1)}$, we solve for each sample's activation vector:

$$\mathbf{v}_r^{(t+1)} = \arg\min_{\mathbf{v}\geq 0} \|\mathbf{z}_r - \mathbf{U}^{(t+1)}\mathbf{v}\|_2^2 + \lambda\|\mathbf{v}\|_1. \quad (10)$$

This non-negative sparse coding problem identifies which fuzzy rules fire for each sample, encouraging interpretable rule selection through sparsity.

### 4.5. Multimodal Concept Grounding

To establish semantic meaning for the learned neuro-fuzzy concepts $\mathbf{U}^*$, we ground each concept vector $\mathbf{u}_j$ ($j \in \{1,\dots,J\}$) in both visual and textual domains, completing the interpretability pipeline from abstract rule representations to concrete semantic content.

**Visual Grounding.** Following prototyping approaches (Kim et al., 2018; Alvarez Melis & Jaakkola, 2018), we identify images that most strongly activate concept $\mathbf{u}_j$ by examining activation coefficients in $\mathbf{V}^*$. Given a visualization budget $N_{\text{MAS}}$, the Maximally Activating Samples (MAS) are: $\mathcal{X}_{j,\text{MAS}} = \arg\max_{\substack{\hat{\mathcal{X}}\subset\mathcal{X} \\ |\hat{\mathcal{X}}|=N_{\text{MAS}}}} \sum_{X\in\hat{\mathcal{X}}} |v_j(X)|$. This reveals the visual patterns to which fuzzy rule $j$ responds, enabling interpretation of its activation conditions through concrete examples.

**Textual Grounding.** Since concept vectors reside in the language model's representation space, we decode them using the unembedding matrix $W_U$. We compute logits $l = W_U\mathbf{u}_j$ (Langedijk et al., 2024; Sakarvadia et al., 2023; Parekh et al., 2024) and extract high-probability tokens, filtering for meaningful English non-stop-words exceeding three characters (Parekh et al., 2024), yielding textual grounding set $\mathcal{T}_j$. These decoded words provide linguistic descriptors for the semantic concept encoded in the rule's consequent. Together, $\mathcal{X}_{j,\text{MAS}}$ and $\mathcal{T}_j$ provide multimodal grounding: visual exemplars show what activates the rule (IF part), while textual descriptors name the semantic concept (THEN part). Figure 2 illustrates this for the "Train" and "Bus" concept.

**Local Interpretation.** To understand how the LMM represents a test image $X$ where token $t$ is predicted, we extract representation $\mathbf{z}_X$ and compute activation profile $\mathbf{v}(X) \in \mathbb{R}^J$. The most activating concepts are: $\tilde{\mathbf{u}}(X) = \{\mathbf{u}_j \in \mathbf{U}^* \mid j \in \text{top-}r \text{ by } |v_j(X)|\}$, identifying which fuzzy rules have the strongest firing strengths. By examining the multimodal grounding of $\tilde{\mathbf{u}}(X)$, we obtain interpretable explanations: which semantic concepts are active,

'train', 'steam', 'engine', 'passenger', 'rail'

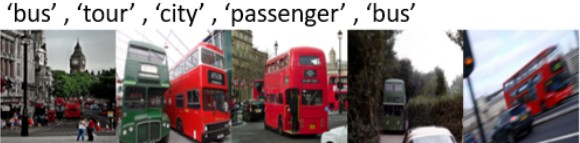

*(a)* The five top-activating visual samples from the decomposed representation $Z$ and the five most probable decoded words for each concept for the token "Train".

'bus', 'tour', 'city', 'passenger', 'bus'

*(b)* The five top-activating visual samples from the decomposed representation $Z$ and the five most probable decoded words for each concept for the token "Bus".

*Figure 2.* Neuro-fuzzy concept grounding.

their firing strengths, and what they mean visually and linguistically. This exposes the rule-based reasoning structure underlying the model's representation of the input, fulfilling our interpretability objective.

## 5. Experiments

### 5.1. Implementation Details

To validate our proposed work, experiments using the **LLaVA** (Liu et al., 2023) and **Qwen2-VL** (Wang et al., 2024a) architectures, evaluated on the COCO dataset (Lin et al., 2014). The LLaVA model incorporates a frozen ViT-L/14 CLIP encoder (Radford et al., 2021) as the visual frontend $f_V$, connected to the language model via an MLP projection. Similarly, Qwen2-VL utilizes a specialized visual encoder coupled with the Qwen2-7B LLM. For uniformity, all results reported in the paper are derived using a fuzzy rule count of $J = 20$. We extract token representations from layer $L = 31$, corresponding to the final transformer layer before the unembedding head. For the proposed Neuro-FeX objective, we set the regularization weight $\lambda = 10^{-2}$ and $\epsilon = 10^{-8}$. Visual grounding is visualized using the 5 most activating samples in $\mathcal{X}_{j,\text{MAS}}$ for any given concept $u_j$. To construct the concept set $\mathcal{T}_j$, we select the 15 tokens with the largest logits and subsequently apply the aforementioned filtration steps as mentioned in 4.5. The data is organized according to the Karpathy split, yielding a training corpus of approximately 120,000 images. Both the validation and test sets contain 5,000 distinct images, each annotated with five independent captions.

## 5.2. Experimental Analysis and Models

We assess the quality of the learned neuro-fuzzy concepts along three complementary dimensions: (i) their effectiveness in interpreting LMM representations at inference time for unseen samples; (ii) the degree of overlap or entanglement among the grounded words associated with different concepts; and (iii) the quality of visual and textual grounding, which supports understanding of the concepts themselves. We consider variants of Neuro-FeX that differ in their neuro-fuzzy rule-generating strategies, including SVD, K-Means, and Gaussian membership functions. To assess concept extraction on test data using CLIPScore and BERTScore, we compare against the following: (i) **CoX-LMM:** A Concept-Based Explainability Framework for LMMs (Parekh et al., 2024). (ii) **Neuro-FeX-v0:** Neuro-FeX with fuzzy-rule centers are obtained by top singular vectors and utilize fuzzy cosine similarity as membership function. (iii) **Neuro-FeX-v1:** Neuro-FeX with fuzzy-rule centers is obtained by K-means clustering on learned representations and utilize a Gaussian fuzzy membership function. (iv) **Neuro-FeX-v2:** Neuro-FeX with fuzzy-rule centers is obtained by K-means clustering and utilize a Gaussian fuzzy membership function. Further, overlap and entanglement within the concept dictionary are compared across PCA, K-Means, and Semi-NMF variants of CoX-LMM. For evaluating visual text grounding, we further compare against the Rnd-Words baseline while keeping the set of maximally activating samples $X_{j,\text{MAS}}$ fixed across methods.

## 5.3. Quantitative Analysis: Multimodal Grounding of Concepts.

### 5.3.1. LEARNED NEURO-FUZZY CONCEPTS DURING INFERENCE

To evaluate how the learned concept dictionary $U^*$ supports interpretation of a test sample $X$, we first identify the top-$r$ most strongly activated concepts, denoted by $\tilde{u}(X)$ (see 4.5). For each selected concept $u_j \in \tilde{u}(X)$, we then assess the correspondence between the test image $X$ and its associated grounded words $T_k$. This correspondence is measured using the average CLIPScore between $X$ and $T_j$, which directly quantifies the alignment between the image embedding and the textual grounding of the most activated concepts. Use of CLIPScore is inspired by (Schwettmann et al., 2023), which helps to validate the multimodal nature of the concept dictionaries.

We evaluate the generality of the proposed Neuro-FeX on LLaVA (Liu et al., 2023), a widely used open-source large multimodal model. LLaVA employs a CLIP ViT-L/336px visual encoder $f_V$, followed by a two-layer linear connector $C$ that produces $N_V = 576$ visual tokens, and a Vicuna-7B language model $f_{LM}$ with 32 layers. The hyperparameters to evaluate the performance are kept similar to the baseline

*Table 1.* **Concept extraction on LLaVA** Mean CLIPScore on the test set for the top-1 activated concept, evaluated on the baseline and the proposed models. Higher values indicate better performance.

| Method | Dog | Cat | Train | Bus |
|---|---|---|---|---|
| CoX-LMM | $\underline{0.618 \pm 0.053}$ | $0.649 \pm 0.046$ | $0.655 \pm 0.066$ | $0.621 \pm 0.076$ |
| Neuro-FeX-v0 | $\mathbf{0.629 \pm 0.047}$ | $0.593 \pm 0.068$ | $0.663 \pm 0.042$ | $0.663 \pm 0.050$ |
| Neuro-FeX-v1 | $0.617 \pm 0.055$ | $\mathbf{0.658 \pm 0.038}$ | $\mathbf{0.667 \pm 0.046}$ | $\underline{0.664 \pm 0.038}$ |
| Neuro-FeX-v2 | $0.617 \pm 0.055$ | $\underline{0.650 \pm 0.038}$ | $\underline{0.664 \pm 0.038}$ | $\mathbf{0.669 \pm 0.046}$ |

(Parekh et al., 2024) as ($J = 20$, $\lambda = 1$, $L = 31$). We report test CLIPScore values for the top-1 activated neuro-fuzzy concept under different baselines, in Table 1. Clearly from the Table 1, the proposed Neuro-FeX surpass CLIPScore for each token. This represents better alignment between the image and text embeddings. For example, in the case of the "Bus" token, Neuro-FeX achieves a CLIPScore of 0.667 (vs 0.621 for CoX-LMM). This suggests that the neuro-fuzzy gating effectively filters out noise in the activation space, leading to concept vectors that align more closely with the visual features of the input images.

**Generalization across diverse token types.** To validate that Neuro-FeX generalizes beyond the initial four concrete object tokens, we conduct additional experiments on a larger and more diverse set of 30 tokens, ensuring comprehensive linguistic coverage across nouns, verbs, adjectives, and abstract concepts (e.g., *wooden, orange, bird, cow, umbrella, backpack, sitting, eating, flying, standing, dark, bright, small, inside, together, behind, holding*). We evaluate on Qwen2.5-VL (Hui et al., 2024) and Qwen3-VL (Yang et al., 2025), reporting mean CLIPScore in Table 2. Neuro-FeX maintains consistently strong performance across all linguistic categories, confirming that the neuro-fuzzy decomposition framework generalizes well beyond concrete object nouns.

*Table 2.* **Generalization to 30 diverse tokens.** Mean CLIPScore comparing Neuro-FeX variants against CoX-LMM on Qwen2.5-VL and Qwen3-VL. Higher is better.

| Model | Neuro-FeX-v0 | Neuro-FeX-v1 | Neuro-FeX-v2 | CoX-LMM |
|---|---|---|---|---|
| Qwen-2.5-VL | 0.621 | **0.643** | 0.624 | 0.616 |
| Qwen-3-VL | 0.590 | **0.592** | 0.587 | 0.582 |

### 5.3.2. ENTANGLEMENT OF LEARNED CONCEPTS

Ideally, each concept in the dictionary $U^*$ should capture distinct information about the token of interest $t$. Consequently, different concepts $u_j$ and $u_l$, with $j \neq l$, should be associated with disjoint or minimally overlapping sets of grounded words. To quantify the degree of entanglement among learned concepts, we measure the overlap between their corresponding word sets $T_j$ and $T_l$.

The overlap of a concept $u_j$ is defined as the average fraction of words it shares with all other con-

*Table 3.* **Overlapping Score on LLaVA.** Overlap evaluation (LLaVA). Lower is better.

| Method | Dog | Cat | Train | Bus |
|---|---|---|---|---|
| PCA | **0.008** | **0.010** | **0.024** | **0.010** |
| K-means | 0.429 | 0.554 | 0.382 | 0.518 |
| CoX-LMM | 0.146 | 0.182 | 0.096 | 0.196 |
| Neuro-FeX-v0 | 0.067 | 0.089 | 0.062 | 0.181 |
| Neuro-FeX-v1 | 0.430 | 0.317 | 0.361 | 0.355 |
| Neuro-FeX-v2 | 0.430 | 0.317 | 0.355 | 0.361 |

*Table 4.* **MonoSemanticity Score (MS)** across three vision backbones ($K=1024$ for all methods, range $[0, 1]$, higher is better). Best results are **bold**, second best are underlined.

| Backbone | Method | Mean MS $\pm$ std | Max MS |
|---|---|---|---|
| CLIP ViT-L | **Neuro-FeX (Ours)** | **0.188 $\pm$ 0.18** | **0.860** |
| | Matryoshka SAE | 0.160 $\pm$ 0.17 | 0.820 |
| | BatchTopK SAE | 0.120 $\pm$ 0.11 | 0.570 |
| | CoX-LMM | 0.102 $\pm$ 0.05 | 0.695 |
| SigLIP SoViT-400m | **Neuro-FeX (Ours)** | **0.711 $\pm$ 0.06** | **0.923** |
| | Matryoshka SAE | 0.610 $\pm$ 0.09 | 0.899 |
| | BatchTopK SAE | 0.610 $\pm$ 0.09 | 0.879 |
| | CoX-LMM | 0.589 $\pm$ 0.05 | 0.744 |
| Qwen3 | **Neuro-FeX (Ours)** | 0.544 $\pm$ 0.06 | **0.918** |
| | Matryoshka SAE | **0.549 $\pm$ 0.10** | 0.889 |
| | BatchTopK SAE | 0.528 $\pm$ 0.10 | 0.845 |
| | CoX-LMM | 0.473 $\pm$ 0.06 | 0.733 |

cepts in the dictionary (Parekh et al., 2024). The overall overlap (or entanglement) of the dictionary $U^*$ is then computed as the average overlap across all concepts: $\text{Overlap}(U^*) = \frac{1}{J} \sum_{j=1}^{J} \text{Overlap}(u_j)$, and $\text{Overlap}(u_j) = \frac{1}{J-1} \sum_{\substack{l=1 \\ l \neq j}}^{J} \frac{|T_l \cap T_j|}{|T_j|}$.

**Concept overlap analysis.** A critical requirement for interpretability is that concepts should be distinct. Table 3 reports the degree of overlap between neuro-fuzzy concepts for the baseline, PCA, K-Means, and CoX-LMM and the proposed variants of Neuro-FeX, using LLaVA. We observe that the K-Means and Neuro-FeX-v1 and Neuro-FeX-v2 exhibit substantially higher overlap between grounded words, indicating a high level of concept entanglement. In contrast, PCA achieves the lowest overlap, with concepts that are nearly disjoint, while Neuro-FeX-v0 shows moderate overlap. Overall, Neuro-FeX-v0 provides the most favourable trade-off: it learns a concept dictionary that is effective for interpreting test image representations while maintaining a high degree of concept diversity and disentanglement. This indicates that Neuro-FeX-v0 achieves the "best of both worlds": the high semantic fidelity of fuzzy clustering with the disentanglement properties of linear factorization.

### 5.3.3. MONOSEMANTICITY OF LEARNED CONCEPTS

Beyond token-specific grounding quality, we evaluate whether Neuro-FeX (v1) learns monosemantic concepts in a general, token-independent setting. We adopt the MonoSemanticity (MS) score (Pach et al., 2025), a metric validated against human perception at $82.8\%$ alignment across 1,000 questions and 71 annotators, which measures whether the top-activating images of a concept are visually coherent with one another.

**Experimental setup.** Following the experimental setup of (Pach et al., 2025), we compare Neuro-FeX directly against BatchTopK SAE (Bussmann et al., 2024) and Matryoshka SAE (Bussmann et al., 2025) with dictionary size $K=1024$ for all methods. Although Neuro-FeX activations $V_{:,k}$ (shaped by RBF membership) and SAE activations (TopK encoder outputs) differ architecturally, both quantify the degree to which each input activates a given concept.

**Results.** Table 4 reports MS across three vision backbones.

Neuro-FeX achieves the highest Max MS across all three encoders, and the highest Mean MS on CLIP ViT-L and SigLIP SoViT-400m. On SigLIP SoViT-400m, Neuro-FeX attains a mean MS of $0.711$, surpassing both SAE baselines by $+16.5\%$. On CLIP ViT-L, mean MS of $0.188$ exceeds Matryoshka SAE ($0.160$, $+17.2\%$). On Qwen3, Neuro-FeX mean MS ($0.544$) is within one standard deviation of Matryoshka SAE ($0.549$, difference $= 0.005$), while achieving superior Max MS ($0.918$ vs $0.889$). Neuro-FeX consistently outperforms BatchTopK SAE and CoX-LMM across all backbones. The consistent improvement reflects a principled architectural advantage: the RBF membership function enforces geometric locality in the representation space, ensuring that concept $k$ activates strongly only for inputs near its center $c_k$. This embeds monosemanticity directly into the activation mechanism, rather than relying on sparsity alone as in SAE-based methods.

Beyond these quantitative evaluations, we provide additional analyses on the proposed Neuro-FeX in the appendix B.1–B.8. Appendix B.1 validates concept grounding against COCO ground-truth human annotations, confirming a positive alignment delta of $+0.036$ above a random-group baseline. Appendix B.2 examines cross-token concept sharing, demonstrating that independently learned per-token dictionaries implicitly reveal linguistic structure across the LMM's representation space. Appendix B.3 validates the faithfulness of the learned IF-THEN rules, independently confirming that fuzzy antecedents partition the representation space semantically and that firing strengths faithfully identify the concepts driving model predictions. Appendix B.4 provides an ablation study isolating the contribution of RBF initialization and fuzzy gating, confirming that RBF initialization reduces reconstruction error by $10.3\times$ compared to random initialization.

'train' , 'passenger' , 'electric' , 'met' , 'colored'

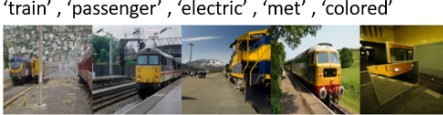

'train' , 'black' , 'passenger' , 'large' , 'rail'

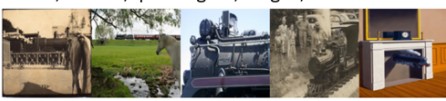

'train' , 'steam' , 'engine' , 'passenger' , 'rail'

'train' , 'steam' , 'engine' , 'antique' , 'model'

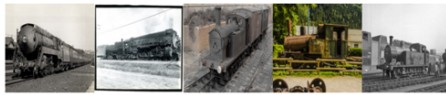

*Figure 3.* Visual–text grounding of concepts. Shown are 4 out of 20 concepts corresponding to the "Train" token (layer 31), along with their five highest-activating visual samples and five most probable decoded words. Concepts 1-4 are shown from top to bottom.

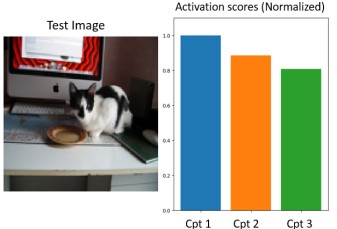
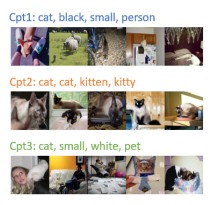

*(a)* Neuro-fuzzy concepts and activation obtained for cat.

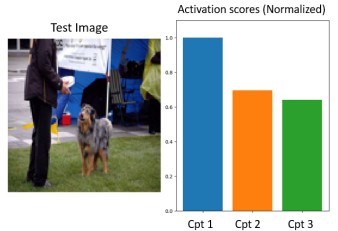
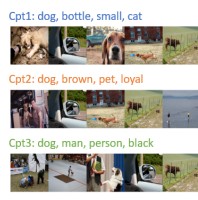

*(b)* Neuro-fuzzy concepts and activation for dog.

*Figure 4.* Local explanations for test examples highlighting different tokens ("Dog", "Cat") obtained using Neuro-FeX at layer 31. We visualize the visual–textual grounding corresponding to the top three normalized concept activations.

## 5.4. Qualitative Results

To assess the quality of visual and textual grounding for a concept $u_k$, we evaluate the correspondence between its associated visual instances $X_{k,\text{MAS}}$ (i.e., the maximally activating samples) and grounded words $T_k$. This alignment is quantified using CLIPScore, as already discussed above.

Figure 3 illustrates the visual and textual grounding of concepts extracted using the proposed Neuro-FeX for the *Train* token. For clarity, we present 4 of the 20 neuro-fuzzy concepts. The neuro-fuzzy concepts associated with the *train* token provide insight into the global structure of the LMM's learned representations. As illustrated in Figure 3, the model captures multiple semantic aspects of the *train* concept. Concepts (1) and (2) correspond to power-related attributes, reflected by grounded terms such as *electric*, as well as physical properties including *large* and *black*. Concepts (3) and (4) encode technical characteristics, such as *steam* and *engine*. Additional concepts describe functional and operational aspects, for example *passenger*. These semantic properties are consistently reflected in both the visual examples and the associated grounded words, demonstrating coherent multimodal grounding.

Figure 4 demonstrates how concept dictionaries learned via Semi-NMF can be used to interpret test-sample representations for the *dog* and *cat* tokens. For each sample, we visualize the normalized activations of the three most strongly activated concepts along with their corresponding multimodal grounding. The most activated concepts

typically capture meaningful and complementary semantic attributes of the input. For example, in the first sample containing a *cat*, concepts corresponding to *pet*, *small/kitty*, and *black/white coloration* are simultaneously activated. The multimodal grounding for the first two concepts is also illustrated in Figure 3.

## 6. Conclusion

In this paper, we introduced Neuro-FeX, a unified interpretability framework for LLMs that explains internal representations through a brain-inspired neuro-fuzzy architecture. By reformulating concept learning as a TSK inference problem, our approach moves beyond purely algebraic decompositions and enables the extraction of explicit, logic-based reasoning rules via fuzzification and defuzzification. The resulting neuro-fuzzy concepts provide semantically grounded interpretations across both vision and language modalities while exposing the underlying "IF-THEN" rule structure that governs model predictions. Through comprehensive qualitative analyses and quantitative evaluations, we demonstrated that these concepts yield meaningful and locally faithful explanations of multimodal representations. The source code is available at https://github.com/mtanveer1/Neuro-FeX-LMM.

## Acknowledgement

Ritik Mishra acknowledges the financial support provided by the Indian Institute of Technology Indore through a research fellowship. M. Sajid acknowledges the Council of Scientific and Industrial Research (CSIR), New Delhi, India, for providing a research fellowship under Grant No. 09/1022(13847)/2022-EMR-I. The authors also thank the Indian Institute of Technology Indore for providing the computational resources, facilities, and research infrastructure that supported this work.

## Impact Statement

As large multimodal models are increasingly deployed in domains where transparency, accountability, and trust are critical, understanding the reasoning behind their predictions has become a fundamental challenge. This work contributes to the development of more interpretable artificial intelligence by introducing Neuro-FeX, a neuro-fuzzy framework that transforms opaque multimodal representations into semantically grounded concepts and explicit fuzzy IF–THEN reasoning rules. Unlike conventional explanation methods that merely identify important input regions, our approach exposes how concepts interact to influence model behavior, providing a human-understandable reasoning structure.

By enabling concept-level auditing and rule-based inspection of large multimodal models, Neuro-FeX has the potential to improve reliability, transparency, and user trust in AI systems deployed in high-stakes applications such as healthcare, education, scientific discovery, and decision support. Furthermore, the proposed framework bridges symbolic reasoning and modern foundation models, opening new research directions toward inherently interpretable and accountable AI systems. While our method does not introduce new generative capabilities and inherits the limitations of the underlying pretrained models, we believe that improving transparency and interpretability represents an important step toward the responsible development and deployment of advanced AI technologies.

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

# A. Appendix: Related Work

## A.1. Concept-Based Explainability

The introduction of CAVs marked a fundamental shift in neural network interpretability (Kim et al., 2018). CAVs represent high-level concepts within a model's activation space by training a linear classifier to distinguish between user-provided concept examples and random counter-examples. The normal vector of the resulting decision boundary defines the CAV, capturing the concept's direction in the latent space. This approach has become foundational for post-hoc interpretability research (Fel et al., 2023a; Gandelsman et al., 2024). Building upon CAVs, subsequent methods have focused on automating concept discovery. ACE (Automated Concept-based Explanation) (Ghorbani et al., 2019) eliminates the need for manual concept curation by automatically extracting visual concepts from model activations. CRAFT (Concept Recursive Activation Factorization) (Fel et al., 2023c) extends this by applying Non-Negative Matrix Factorization (NMF) to decompose image patch activations into class-specific concepts. More recently, the CCR framework (Liang et al., 2025) improves concept embedding stability through bounded optimization, enabling precise control over the accuracy-explainability trade-off. Despite these advances, concept-based explainability methods have primarily targeted convolutional neural networks for classification tasks. Large Multimodal Models (LMMs), which integrate vision and language through transformer architectures, present fundamentally different interpretability challenges. Recent work has begun exploring concept extraction in multimodal settings (Parekh et al., 2024), yet the application of interpretable reasoning structures to LMMs remains largely unexplored. We address this gap by introducing a neuro-fuzzy framework that combines concept-based explainability with brain-inspired fuzzy logic for interpretable multimodal reasoning.

## A.2. Large Multimodal Models

LLMs (Brown et al., 2020; Touvron et al., 2023) have emerged as the backbone of modern LMMs. The typical LMM architecture comprises three core components: a pre-trained LLM that provides language understanding and generation capabilities, a visual encoder (often based on vision transformers) that processes image inputs, and a connector module that bridges the two modalities. This modular design enables LMMs to leverage the semantic knowledge encoded in pre-trained LLMs while extending their capabilities to visual reasoning. Recent LMMs have demonstrated strong performance across diverse multimodal tasks, including visual question answering (VQA) (Kuang et al., 2025), image captioning (Chu et al., 2025), and visual commonsense reasoning (Babaiee et al., 2025). Models such as Flamingo (Alayrac et al., 2022), BLIP-2 (Li et al., 2023), and LLaVA (Liu et al., 2023) exemplify this paradigm, achieving impressive results by keeping the LLM and visual encoder frozen while training only the lightweight connector module. This training strategy raises intriguing questions about how frozen LLMs generalize to multimodal inputs and what internal representations emerge during multimodal processing. Despite these achievements, the black-box nature of LMMs poses significant challenges for deployment in high-stakes applications. Understanding how these models integrate visual and linguistic information, what multimodal concepts they learn, and how they reason over combined inputs remains an open problem. This motivates the need for explainability methods specifically designed for the unique architecture and processing characteristics of LMMs.

## A.3. Neuro-Fuzzy Systems

Neuro-fuzzy systems combine the learning capabilities of neural networks with the interpretable reasoning structure of fuzzy logic systems (de Campos Souza, 2020). These hybrid models employ fuzzy IF–THEN rules with linguistic variables, enabling human-comprehensible decision-making processes while maintaining the adaptive power of neural architectures. The TSK model represents a particularly influential neuro-fuzzy framework, wherein rule consequents are defined as linear functions of inputs rather than fuzzy sets, facilitating efficient defuzzification and gradient-based optimization. Recent applications of neuro-fuzzy systems have focused on biomedical domains (Tanveer et al., 2024; Wang et al., 2024b), control systems, time-series prediction (Ding et al., 2026), robotics (Li et al., 2026), and in the security of autonomous vehicle systems (Cai et al., 2026). These systems are valued for their dual interpretability: the fuzzy rule base provides transparent logical reasoning, while the neural learning component enables data-driven adaptation. However, neuro-fuzzy approaches have not been extensively applied to modern deep learning architectures, particularly in the context of interpretability for large-scale multimodal models. We bridge this gap by adapting neuro-fuzzy principles to the interpretability of LMMs. Rather than training a neuro-fuzzy system from scratch, we interpret the latent representations of pre-trained LMMs to expose an implicit fuzzy inference structure, thereby combining the performance of end-to-end learned models with the interpretability of rule-based reasoning.

## B. Appendix: Additional Experimental Analysis

### B.1. Caption-Concept Alignment

The following analysis is conducted on Qwen2-VL-7B, extracting token-specific hidden states from the `model.norm` layer ($L=28$, $D=3584$), following the implementation details of Section 5.1.

To validate concept grounding quality against human-authored references, we evaluate alignment between automatically extracted grounding words $T_j$ and COCO ground-truth captions (5 captions per image, crowdsourced via Amazon Mechanical Turk) (Lin et al., 2014). For each concept $j$, we compute the mean CLIP cosine similarity between $T_j$ and the GT captions of its top-5 MAS images, benchmarked against 100 random-group baselines of equal size. Table 5 reports results across random 5 tokens.

*Table 5.* **Caption-concept alignment** using COCO ground-truth human annotations. Delta above random = alignment minus random-group baseline. Higher is better.

| Metric | Neuro-FeX |
|---|---|
| Caption-concept alignment | $0.079 \pm 0.045$ |
| Delta above random | $+\mathbf{0.036 \pm 0.021}$ |
| Fraction of concepts > random | $\mathbf{70}\%$ |

Neuro-FeX achieves a positive delta of $+0.036$ above the random baseline, with 70% of concepts individually exceeding it. This confirms that the grounding words $T_j$ extracted by Neuro-FeX carry genuine semantic alignment with human-authored image descriptions, providing evidence of human-grounded interpretability without requiring a dedicated user study.

### B.2. Cross-Token Concept Sharing

Although Neuro-FeX operates token-specifically by design, the independently learned concept dictionaries $U_t$ all reside in the shared ambient representation space of the underlying LMM. We investigate whether this independence conceals latent cross-token structure by applying Neuro-FeX to five tokens spanning three linguistic categories (noun: *dog*; adjectives: *orange*, *wooden*; verbs: *jumping*, *riding*), learning $K=20$ concept vectors per token. Shared concepts between any two dictionaries $U_t$ and $U_{t'}$ are identified via Hungarian matching (Kuhn, 1955), finding the optimal 1-to-1 concept pairing by cosine similarity. Table 6 reports mean matching similarity for all token pairs.

*Table 6.* **Cross-token concept similarity** via Hungarian matching. Each entry is the mean cosine similarity of matched concept pairs between $U_t$ and $U_{t'}$. Same-type pairs share substantially more structure than cross-type pairs.

| | dog | orange | wooden | jumping | riding |
|---|---|---|---|---|---|
| **dog** | 1.000 | 0.454 | 0.465 | 0.176 | 0.253 |
| **orange** | 0.454 | 1.000 | 0.502 | 0.171 | 0.220 |
| **wooden** | 0.465 | 0.502 | 1.000 | 0.158 | 0.231 |
| **jumping** | 0.176 | 0.171 | 0.158 | 1.000 | 0.352 |
| **riding** | 0.253 | 0.220 | 0.231 | 0.352 | 1.000 |

As demonstrated in Table 6, same-type token pairs share significantly more concepts than cross-type pairs: verb-verb (0.352) and adjective-adjective (0.502) both substantially exceed the cross-type mean (0.271). This reveals that the LMM internally organises representations along linguistic category lines, a structure that Neuro-FeX surfaces through independent per-token decompositions without any cross-token supervision.

### B.3. Faithfulness of Learned IF-THEN Rules

We validate that the fuzzy IF-THEN rules learned by Neuro-FeX faithfully capture the model's internal reasoning by independently evaluating the antecedent (IF part) and consequent (THEN part) of each rule across five tokens on Qwen2-VL-7B.

**Rule Consistency (IF part).** The Gaussian membership function assigns firing strength based on geometric proximity to concept center $c_k$: if these antecedents are meaningful, the top-activating images of each rule should form visually coherent

groups. For each rule, we retrieve the top-5 Maximally Activating Samples (MAS) and compute their mean pairwise CLIP coherence, benchmarked against 100 random groups of equal size from the same dataset.

Table 7 shows that Neuro-FeX achieves ConsDelta=+0.119, with 94% of rules (19/20) individually exceeding the random baseline. This confirms that the fuzzy membership functions partition the representation space along semantically meaningful boundaries rather than arbitrarily.

**Faithfulness (THEN part).** If the firing strengths genuinely reflect the model's reasoning, selecting the top-$r$ rules should reconstruct the hidden state more faithfully than selecting $r$ random rules. Following (Samek et al., 2021), we measure target token prediction quality under $r$-concept reconstruction and define the faith gap as the difference in LogitRatio between top-$r$ and random-$r$ selections.

As shown in Table 8, top-$r$ rules carry $4.36\times$ more prediction signal than random at $r{=}1$, and $2.81\times$ at $r{=}5$. This directly confirms that firing strength rankings identify the rules that drive model predictions, validating the THEN consequents as faithful representations of internal reasoning.

*Table 7.* **Rule consistency** (mean $\pm$ std, 5 tokens). ConsDelta = CLIP coherence of top-5 MAS minus random baseline.

| Metric | Neuro-FeX | Random |
|---|---|---|
| Mean CLIP coherence | $0.651 \pm 0.029$ | $0.532 \pm 0.021$ |
| ConsDelta | $+\mathbf{0.119}$ | - |
| Rules > random | $\mathbf{94}\% \, (19/20)$ | - |

*Table 8.* **Faithfulness** (mean $\pm$ std, 5 tokens). Faith gap = top-$r$ minus random-$r$ LogitRatio.

| Top-$r$ | Neuro-FeX | Random | Faith gap |
|---|---|---|---|
| $r{=}1$ | $0.266 \pm 0.105$ | $0.061 \pm 0.019$ | $+0.205 \, (4.36\times)$ |
| $r{=}5$ | $0.826 \pm 0.107$ | $0.294 \pm 0.029$ | $+0.532 \, (2.81\times)$ |
| $r{=}10$ | $1.046 \pm 0.029$ | $0.547 \pm 0.061$ | $+0.499 \, (1.91\times)$ |

Together, these results confirm that Neuro-FeX's fuzzy IF-THEN rules are not a representational convenience: the antecedents impose semantically grounded geometric conditions on the representation space, and the consequents produce firing strengths that faithfully reflect the model's predictive reasoning.

## B.4. Ablation Study

We ablate the contribution of each component of the Neuro-FeX decomposition across four tokens ($N_{\text{test}} \geq 50$) on Qwen2-VL-7B. Four variants are evaluated: (A) Full Neuro-FeX with RBF initialization and fuzzy gating; (B) random initialization with fuzzy gating; (C) RBF initialization without gating; and (D) random initialization without gating, equivalent to a standard NMF baseline. We report two metrics: reconstruction error $\text{ReconErr} = \|Z - UV\|_F^2 / N$ on the training set (lower is better), and ConsDelta - mean pairwise CLIP coherence of top-5 MAS per rule minus the random-group baseline (higher is better).

*Table 9.* **Ablation study** (mean $\pm$ std, 4 tokens, $N_{\text{test}} \geq 50$). ReconErr $= \|Z - UV\|_F^2 / N$, lower is better. ConsDelta = CLIP coherence of top-5 MAS per rule minus random baseline, higher is better.

| Variant | Description | ReconErr ↓ | ConsDelta ↑ |
|---|---|---|---|
| A | Full Neuro-FeX (RBF init + gating) | $\mathbf{0.43 \pm 0.15}$ | $+0.099 \pm 0.018$ |
| B | Random init + gating | $4.47 \pm 1.52$ | $+0.043 \pm 0.071$ |
| C | RBF init, no gating | $\underline{0.46 \pm 0.14}$ | $\underline{+0.110 \pm 0.028}$ |
| D | Random init, no gating (NMF baseline) | $4.47 \pm 1.52$ | $+0.035 \pm 0.060$ |

*Table 10.* **Component contributions** (Variant A minus ablation variant). $\Delta$ReconErr and $\Delta$ConsDelta measure the gain attributable to each component.

| Comparison | $\Delta$ReconErr | $\Delta$ConsDelta |
|---|---|---|
| A vs B: RBF initialization | $\mathbf{-4.04 \, (10.3\times)}$ | $+\mathbf{0.057 \, (57\%)}$ |
| A vs D: total fuzzy vs NMF | $\mathbf{-4.04 \, (10.3\times)}$ | $+\mathbf{0.065 \, (65\%)}$ |

Replacing RBF initialization with random initialization (Variants B and D) increases reconstruction error by $10.3\times$ (0.43 vs 4.47) and degrades ConsDelta by 57% (+0.099 vs +0.043), confirming that RBF initialization is the dominant structural contribution of the fuzzy membership framework. The fuzzy gating mechanism (A vs C) contributes a complementary

architectural role, enforcing the TSK interpretation by ensuring activations remain geometrically conditioned on membership values throughout inference.

### B.5. Downstream application: concept-level bias detection.

Beyond grounding quality, the learned concepts directly support model auditing. For each target token, we score the top-5 MAS per concept against bias-relevant text prompts via CLIP, computing a concentration score to quantify visual sub-distribution dominance (threshold = 0.4). As reported in Table 11, no concept among 100 total exceeds the threshold, while revealing actionable model insights: *wooden* is furniture-dominant (13/20 concepts), *riding* is child-dominant (9/20), and *orange* is colour-dominant rather than fruit-dominant (10/20). Visual concept grids for all five tokens are provided in Appendix B.6. These findings demonstrate downstream utility that aggregate metrics such as CLIPScore and attribution maps cannot provide, supporting targeted model fine-tuning and bias mitigation.

*Table 11.* **Concept-level bias analysis** ($K$=20 per token, 100 total). Concentration threshold = 0.4.

| Token | Type | Key insight |
|---|---|---|
| dog | noun | Balanced across 8 visual contexts |
| orange | adjective | 10/20 concepts colour-dominant |
| wooden | adjective | 13/20 concepts furniture-dominant |
| jumping | verb | 11/20 concepts person-dominant |
| riding | verb | 9/20 concepts child-dominant |

### B.6. Visual Concept Grids for Bias Detection

Figures 5–9 present the visual concept grids for all five target tokens (*dog*, *orange*, *wooden*, *jumping*, *riding*), showing the top-5 Maximally Activating Samples per concept alongside their concentration scores. For each token, we highlight the five concepts with the highest concentration scores. These grids provide direct visual evidence of the sub-distribution structure identified by the bias detection analysis reported in Table 11 of the main paper, enabling qualitative verification of the quantitative concentration scores.

### B.7. Additional Experiments with Qwen2-VL-7B

#### Quantative analysis on Qwen2-VL-7B

To further assess the generality of our approach, we conduct additional experiments on Qwen2-VL-7B, a widely used open-source large multimodal model (Table 12). The Qwen2-VL-7B model uses a SigLIP-based visual encoder, a 2-layer MLP connector ($C$), and a Qwen2-7B language model ($f_{LM}$, 28 layers). The hyperparameter for experiments on Qwen are $K = 20, \lambda = 10^{-2}, L = 28$. On Qwen2-VL-7B, the proposed method achieves a higher CLIPScore, while the overlap score for the Neuro-FeX-v0 variant remains comparable to competing approaches. These results suggest that the latent space learned by Qwen2 is compatible with the sharper decision boundaries induced by the proposed neuro-fuzzy rules.

**Qualitative analysis on Qwen2-VL-7B** We have also performed qualitative analysis on Qwen2-VL-7B. The qualitative visualizations of concepts extracted from Qwen2-VL using the proposed Neuro-FeX, for the *train* token in Figure 10, and include additional examples for the *cat* and *dog* tokens in Figures 11 and 12.

### B.8. Ablation analysis on hyperparameter $\gamma$ for Gaussian fuzzy membership function

Activation (fire strength) plays an important role in building neuro-fuzzy concepts, as it captures nonlinearity and enables models to represent complex relationships. By shaping the extraction of concepts, Activation governs how decision-making propagates through the network using neuro-fuzzy concepts. Activation directly depends on the Gaussian function being used to find the degree of importance of the generated Neuro-fuzzy concept.

We analyze the quality of multimodal concept grounding across different values of the Gaussian scaling parameter $\gamma$. Figure 13 reports the CLIPScore between $(X_{k,\text{MAS}}, T_k)$ for the *dog* token over the range $\{10^5, 10^4, \ldots, 10^{-5}\}$ on both Qwen2 and LLaVA models. For larger values of $\gamma$, the CLIPScore remains relatively stable with minimal variation. As $\gamma$ decreases,

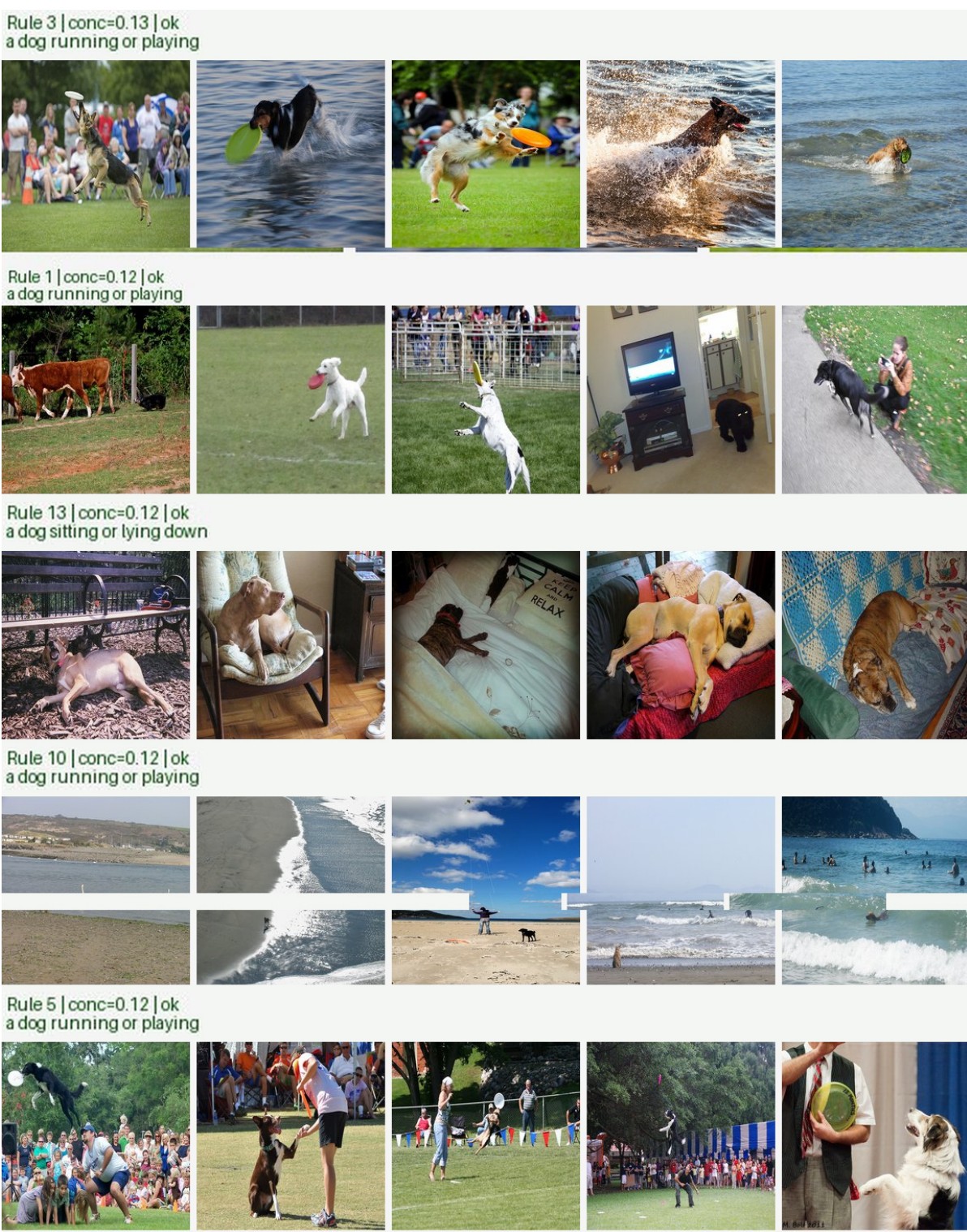

*Figure 5.* **Bias detection - *dog* (noun).** Top-5 concepts by concentration score. The concepts span 8 distinct visual contexts including sitting/lying, running/playing, and with a person, revealing a well-balanced internal representation with no single dominant context.

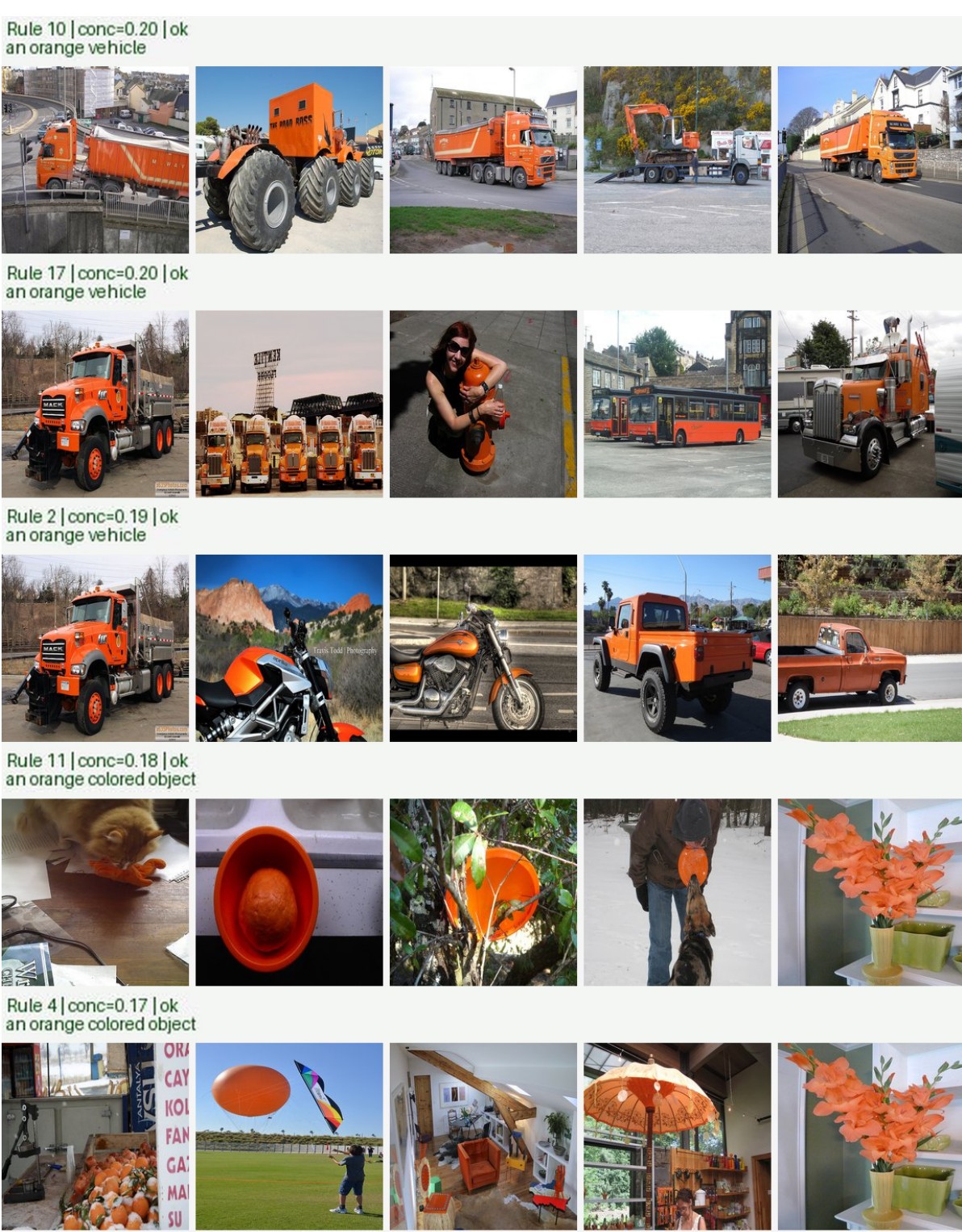

*Figure 6.* **Bias detection - *orange* (adjective).** Top-5 concepts by concentration score. Dominant concepts reflect colour contexts rather than the fruit referent, revealing a colour-dominant internal representation for this adjective.

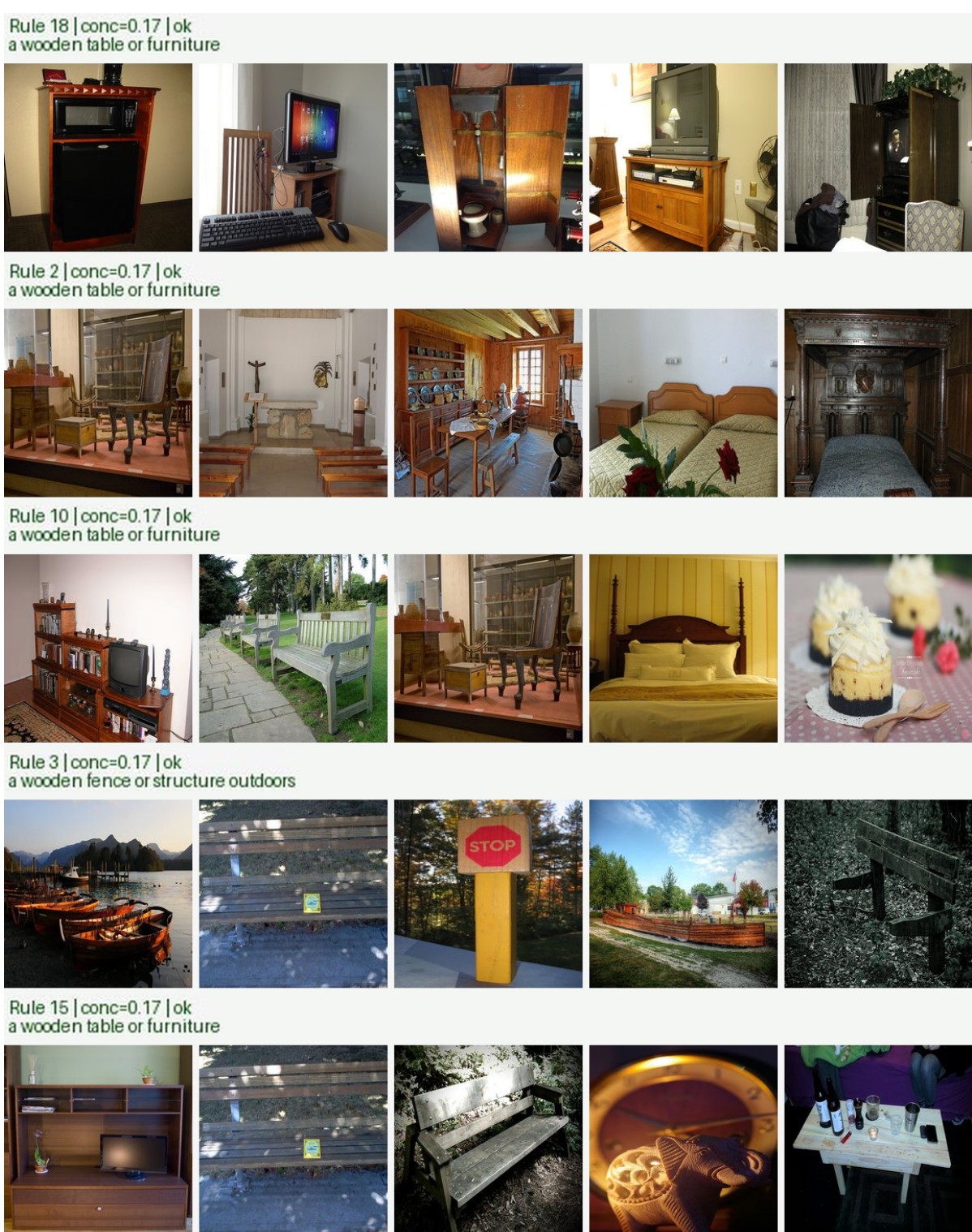

*Figure 7.* **Bias detection - *wooden* (adjective).** Top-5 concepts by concentration score. Dominant concepts reflect furniture contexts (tables, chairs, benches), revealing a furniture-dominant internal representation for this material adjective.

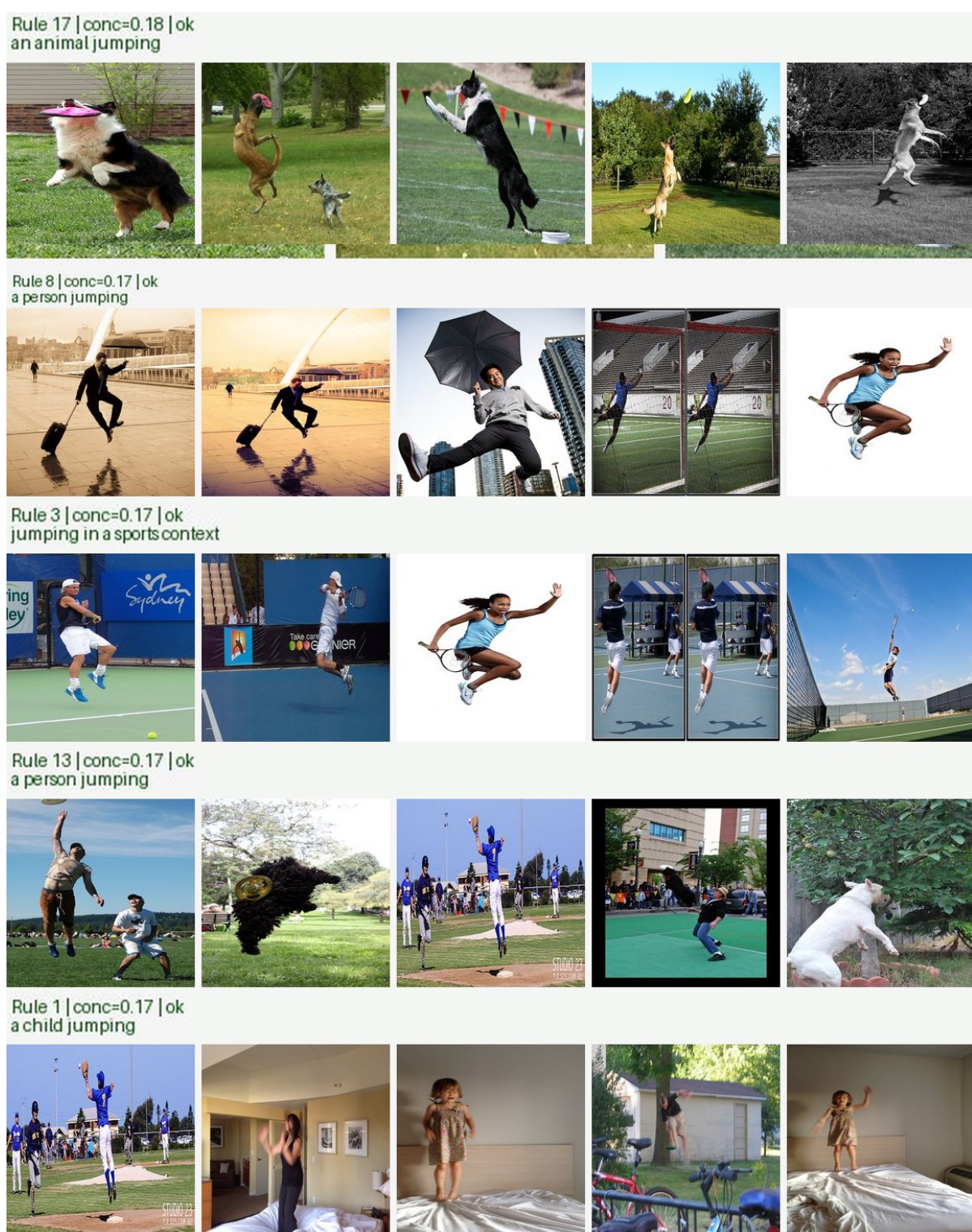

*Figure 8.* **Bias detection - *jumping* (verb).** Top-5 concepts by concentration score. Dominant concepts reflect human jumping contexts, with secondary representation of animal and child jumping, revealing a human-centric bias in the model's representation of this action verb.

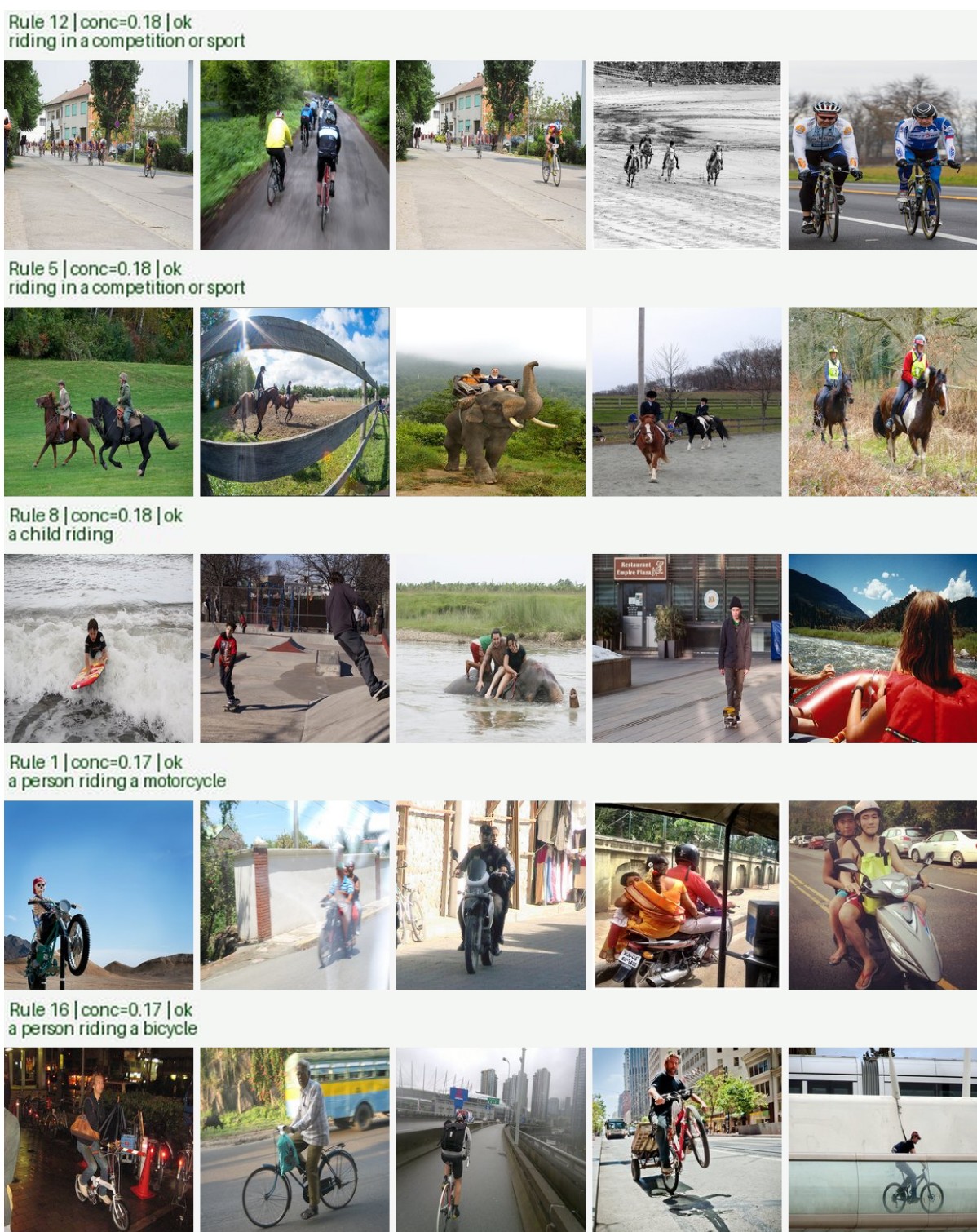

*Figure 9.* **Bias detection - *riding* (verb).** Top-5 concepts by concentration score. Dominant concepts reflect child-riding and competitive sport contexts, revealing a child-centric bias in the model's representation of this relational action verb.

*Table 12.* **Quantitative Evaluation on Qwen2-VL.** We compare Semantic Grounding (CLIPScore ↑) and Disentanglement (Overlap ↓). **NFX-v0** outperforms the baseline in grounding while maintaining competitive sparsity. Best scores are bolded, second best underlined.

| Target Token | Method | CLIPScore ↑ | Overlap ↓ |
|---|---|---|---|
| **Dog** | Semi-NMF (Baseline) | $0.603 \pm 0.067$ | **0.036** |
| | Neuro-FeX-v0 (Ours) | $\mathbf{0.624 \pm 0.046}$ | 0.076 |
| | Neuro-FeX-v1 (Ours) | $0.619 \pm 0.052$ | 0.375 |
| | Neuro-FeX-v2 (Ours) | $\underline{0.621 \pm 0.050}$ | 0.221 |
| Cat | Semi-NMF (Baseline) | $0.628 \pm 0.050$ | **0.027** |
| | **Neuro-FeX-v0 (Ours)** | $0.651 \pm 0.036$ | 0.077 |
| | Neuro-FeX-v1 (Ours) | $\mathbf{0.658 \pm 0.039}$ | 0.494 |
| | Neuro-FeX-v2 (Ours) | $\underline{0.657 \pm 0.043}$ | 0.292 |
| Train | Semi-NMF (Baseline) | $0.644 \pm 0.059$ | **0.049** |
| | **Neuro-FeX-v0 (Ours)** | $\mathbf{0.667 \pm 0.043}$ | 0.081 |
| | Neuro-FeX-v1 (Ours) | $\mathbf{0.653 \pm 0.041}$ | 0.351 |
| | Neuro-FeX-v2 (Ours) | $0.652 \pm 0.050$ | 0.240 |
| **Bus** | Semi-NMF (Baseline) | $0.603 \pm 0.073$ | **0.095** |
| | **Neuro-FeX-v0 (Ours)** | $0.630 \pm 0.075$ | 0.408 |
| | Neuro-FeX-v1 (Ours) | $0.657 \pm 0.044$ | 0.484 |
| | Neuro-FeX-v2 (Ours) | $\mathbf{0.672 \pm 0.046}$ | 0.504 |

grounding quality improves, with CLIPScore increasing and reaching its maximum at lower values. This trend is consistent across both models, indicating that appropriately scaled activation sharpness is critical for effective multimodal concept grounding.

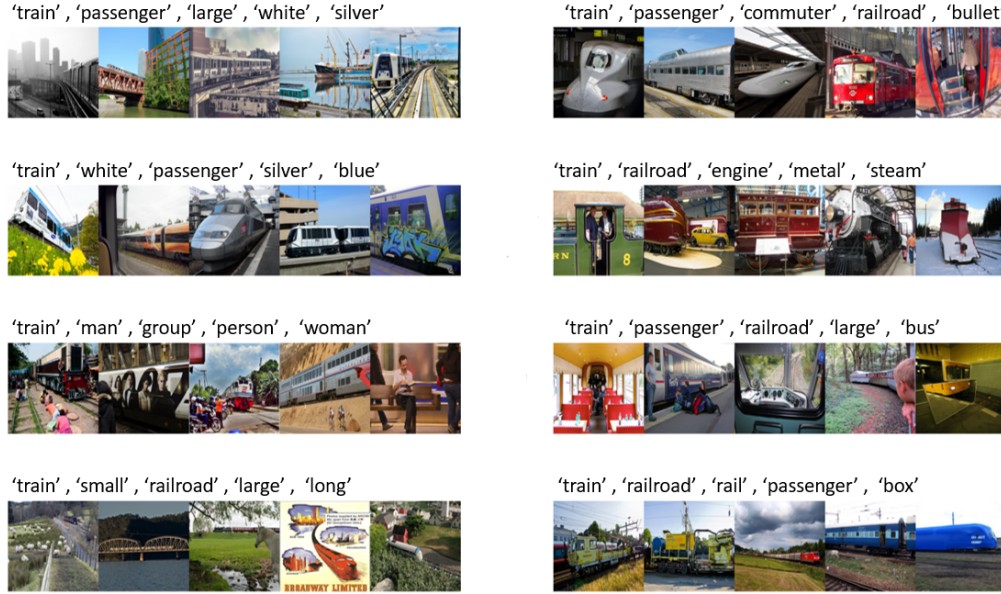

*Figure 10.* Visual–text grounding of concepts on Qwen2-VL-7B. Shown are 8 out of 20 concepts corresponding to the "Train" token, along with their five highest-activating visual samples and five most probable decoded words. Concepts 1-4 are shown on the left, from top to bottom, while concepts 5-8 are displayed on the right.

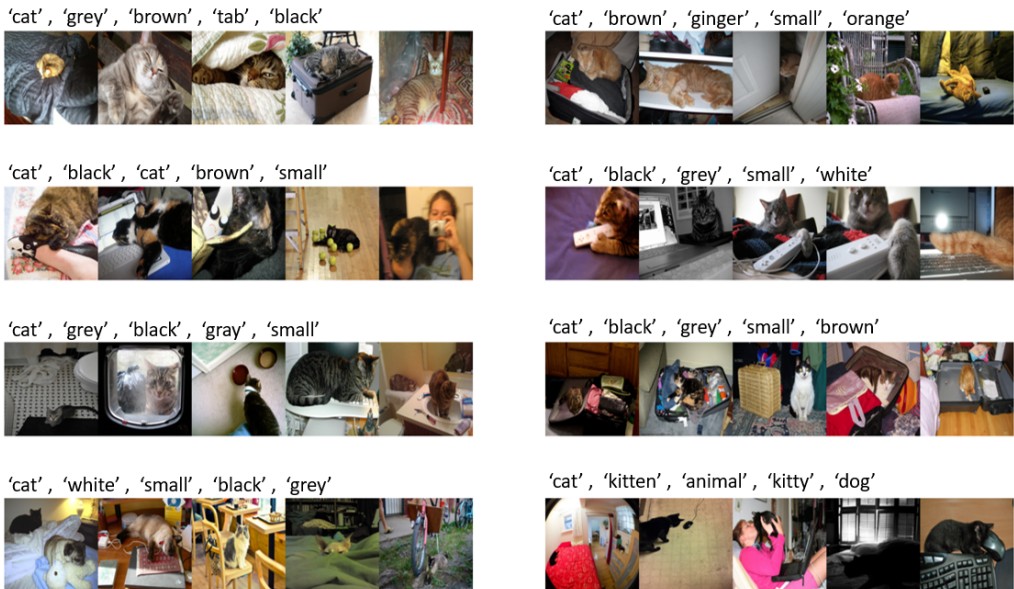

*Figure 11.* Visual–text grounding of concepts on Qwen2-VL-7B. Shown are 8 out of 20 concepts corresponding to the "Cat" token, along with their five highest-activating visual samples and five most probable decoded words.

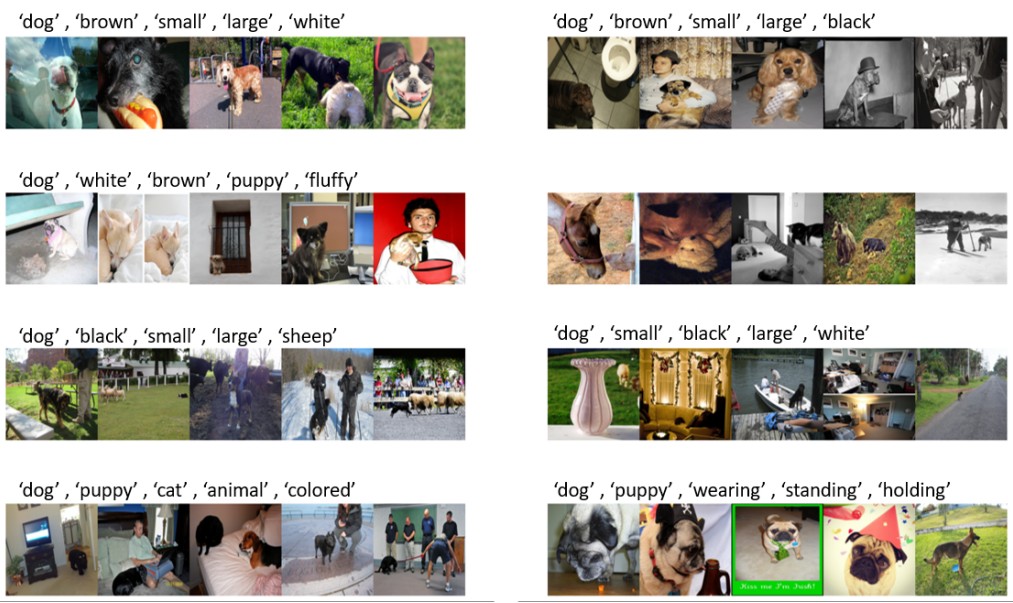

*Figure 12.* Visual–text grounding of concepts on Qwen2-VL-7B. Shown are 8 out of 20 concepts corresponding to the "Dog" token, along with their five highest-activating visual samples and five most probable decoded words.

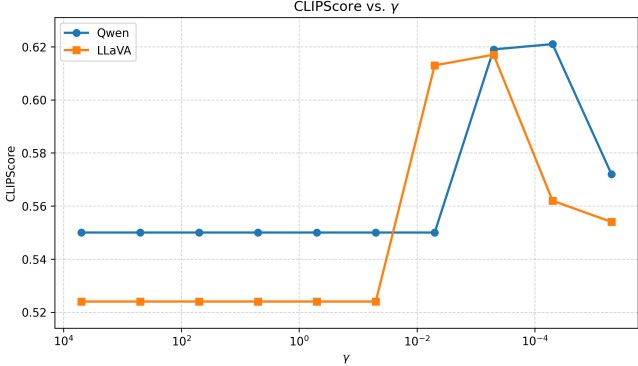

*Figure 13.* CLIPScore between visual and textual grounding $(X_{j,\mathrm{MAS}}, T_j)$. Results are shown for the *dog* token for Qwen and LLaVA model.

