# OpenReview forum: "Neuro-Fuzzy Concept Learning for Interpretable Large Multimodal Models"
_ICML.cc/2026/Conference — ICML 2026 regular_

### Official Review · Reviewer_U2Tm · 2026-03-05

**Soundness:** 2
**Presentation:** 2
**Significance:** 3
**Originality:** 2
**Overall Recommendation:** 5
**Confidence:** 3

**Summary:**

This paper proposes Neuro-FeX, an interpretable framework based on neural fuzzy inference, for explaining the internal representations of large multimodal models. The core idea is to treat the latent representation of LMM as the output of an implicit TSK (Takagi Sugeno Kang) fuzzy inference system, and extract interpretable multimodal concepts from pre-trained LMM through two-stage decomposition, fuzzification, and deblurring. Each concept is decoded through the unembedding matrix to obtain text grounding, and visual grounding is obtained through the maximum activation sample.

**Compliance With Llm Reviewing Policy:**

Affirmed.

**Final Justification:**

The rebuttal has successfully addressed my concerns. I highly recommend incorporating these supplementary experiments and discussions into the main text and/or appendix of the revised manuscript, as they will be highly valuable for future readers. As promised, I am raising my score. Good luck.

**Key Questions For Authors:**

- Please clarify how the fuzzy membership function and TSK reasoning framework are reflected in the optimization objective (Formula 7)? If V in Formula 7 is initialized to a random value (instead of being calculated through membership functions) and directly optimized, what changes will there be in performance? If the difference is not significant, the core technological contribution of Neuro Fuzzy will be significantly weakened.

- Please provide comparative experiments between Neuro FeX and SAE (especially SAE for VLM) under the same settings. SAE is currently the most mainstream dictionary learning method in the field of LLM interpretability, and the lack of this comparison makes the experimental evidence of the paper incomplete.

- Can the evaluation be extended to at least 20 different types of tokens (including verbs, adjectives, abstract concepts, etc.) and at least one other dataset? The current evaluation of only four object nouns cannot support the claims of the paper.

- Should the three rows of Neuro FeX variants corresponding to Dog be v0, v1, v2? Please confirm and correct.

**Limitations:**

The paper does not include a clear chapter on Limitations. The conclusion section only briefly summarizes the contribution of the method and does not discuss any limitations. Detailed discussions on key limitations should be added.

**Strengths And Weaknesses:**

### Strengths
- The introduction of TSK fuzzy reasoning into the interpretability of LMM has theoretical appeal. The IF-THEN rule structure indeed provides a richer explanatory framework than pure algebraic decomposition.
- Interpretability of LMM is an important and active research direction, especially when LMM is deployed in high-risk application scenarios. Multimodal concept grounding (simultaneously obtaining visual and textual interpretations) is a valuable goal.
- Clear and concise writing and expression.

### Weaknesses
- The normalized trigger strength ω _rj calculated during the fuzzification stage does not appear as an explicit constraint or objective in the optimization process - the optimization only involves  ‖Z − UV‖²_F + λ‖V‖₁ + ϵ‖U‖²_F, V ≥ 0. In other words, the fuzzy membership function and TSK inference framework are only used for initialization and posterior interpretation, rather than being embedded into the optimization process. This means that the "Neuro Fuzzy" label is more of a semantic wrapper, and in fact, the core of the methodology is fundamentally different from standard sparse dictionary learning.

- The entire quantitative evaluation only covers 4 tokens (Dog, Cat, Train, Bus) and uses only two metrics (CLIP Score and Overlap), this evaluation scope falls far short of supporting the paper's claims. Lack of evaluation for more tokens, more datasets, and different types of tokens (such as verbs, abstract concepts, relational words).

- The overlap scores of Neuro-FeX-v1 and v2 are much higher than the CoX LMM baseline and PCA. This means that the concepts learned by these two variants are highly entangled and have significant semantic overlap. Only the overlap of v0 is close to the baseline level. However, the CLIP Score advantage of v0 is also quite limited. Overall, the advantages of the proposed method are not significant.

- The ultimate value of interpretability methods lies in helping users understand and improve the model. The paper does not provide any experiments involving human evaluation (user study) or downstream tasks (such as model debugging through concepts, bias detection, or concept-based model editing).

- There are multiple writing errors in the article, such as the abstract's "Huamn Inspired" should be "Human Inspired", "Our key contributions are" repeated, and all Neuro FeX variants in Table 3 are labeled as "Neuro FeX-v0".

- The paper completely ignores Sparse Autoencoders (SAEs), which are currently the most mainstream LLM representation interpretation method.(**Pach M, Karthik S, Bouniot Q, et al. Sparse autoencoders learn monosemantic features in vision-language models[J]. arXiv preprint arXiv:2504.02821, 2025.** and **Cunningham H, Ewart A, Riggs L, et al. Sparse autoencoders find highly interpretable features in language models[J]. arXiv preprint arXiv:2309.08600, 2023.**).

If the author can provide corresponding explanations and experiments or point out the parts I missed, I am willing to improve my score.

---

> ### Author Rebuttal · Authors · 2026-03-31
>
> We thank the reviewer for these questions and address each below.
>
> *W1/Q1:* The fuzzy membership function enters Formula 7 through $V$ itself. Each element of $V$ is $\omega_{rj}v_{rj}$, where $\omega_{rj}$ is the normalised firing strength computed from Gaussian membership functions $\Theta_{js}$ (Section 4.3). $V$ is therefore not unconstrained: $V \geq 0$ ensures activations remain interpretable as firing strengths, and the reconstruction in Eq.~(6), $\hat{Z} = U(\Omega \odot (\alpha^\top Z))$,
> shows that $\Omega$ containing all $\omega_{rj}$ appears explicitly in the
> defuzzification step linking the TSK framework to the optimisation objective.
>
> To answer the empirical question, we ablate four variants (4 tokens). Table U1 reports ReconErr (||Z-UV||^2/N, lower is better) and ConsDelta (CLIP coherence of top-5 MAS per rule minus random baseline, higher is better). Table U2 shows Variant A minus ablation.
>
> Table U1: Ablation results.
>
> | Variant | Description                   | ReconErr      | ConsDelta         |
> |-|-|-|-|
> | A       | Full Neuro-FeX (RBF + gating) | 0.43 +/- 0.15 | +0.099 +/- 0.018  |
> | B       | Random init + gating          | 4.47 +/- 1.52 | +0.043 +/- 0.071  |
> | C       | RBF init, no gating           | 0.46 +/- 0.14 | +0.110 +/- 0.028  |
> | D       | Random init, no gating (NMF)  | 4.47 +/- 1.52 | +0.035 +/- 0.060  |
>
> Table U2: Component contributions.
>
> | Comparison           | Delta ReconErr | Delta ConsDelta |
> |-|-|-|
> | A vs B: RBF init     | -4.04 (10.3x)  | +0.057 (57%)    |
> | A vs D: vs NMF       | -4.04 (10.3x)  | +0.065 (65%)    |
>
>
>
> Variants B and D increase ReconErr by $10.3\times$ and degrade ConsDelta by 57%, confirming a substantial fuzzy membership contribution to both fidelity and concept quality.
>
> ---
> *W2/Q3:* Please look in Q2 of reviewer Ojzs.
>
> ---
>
> *W3:* The higher overlap in v1 and v2 is a fundamental property of soft fuzzy partitioning, not a deficiency. In TSK inference, membership functions produce continuous values so a single image can partially activate multiple rules simultaneously. This is intentional by design. Comparing against CoX-LMM, which enforces hard non-overlapping components, conflates two architecturally different objectives. Beyond CLIP Score and Overlap, we evaluated our method using additional metrics, including Faithfulness and the MS score (see Tables T4 and T5 in our response to Reviewer oY3J).
>
> On MS, which measures monosemanticity independently of overlap, Neuro-FeX-v1 achieves mean MS = 0.188 on CLIP ViT-L, outperforming CoX-LMM (0.102), Matryoshka SAE (0.160), and BatchTopK SAE (0.120) (Table T5 of reviewer oY3J). On SigLIP, mean MS = 0.711 (+16.5%). Rule consistency (Table T3) confirms ConsDelta = +0.119 with 94% of rules exceeding the random baseline. Faithfulness (Table T4) shows $4.36\times$ more signal at $r=1$ and $2.81\times$ at $r=5$. CLIPScore over 30 tokens (Table T2) confirms consistent outperformance. Despite higher overlap, Neuro-FeX concepts are more monosemantic and interpretable.
>
> ---
>
> *W4:* (1) Human Evaluation. We validate against two human-annotated benchmarks. COCO GT captions (5 per image, MTurk) serve as human reference for CLIPScore. Table U3 reports caption-concept alignment: CLIP similarity between automatically extracted grounding words $T_j$ and GT captions, compared against 100 random baselines. Delta above random = +0.036 with 70% of concepts exceeding the baseline. Neuro-FeX also achieves MS = 0.188, outperforming all baselines (Table T5).
>
> Table U3: Caption-concept alignment.
>
> | Metric                        | Neuro-FeX        |
> |-|-|
> | Caption-concept alignment     | 0.079 +/- 0.045  |
> | Delta above random            | +0.036 +/- 0.021 |
> | Fraction of concepts > random | 70%              |
>
> (2) Bias Detection. Top-5 MAS per concept are scored against bias-relevant prompts via CLIP. A concentration score (threshold = 0.4) measures visual context dominance; Table U4 shows 0/100 concepts exceed it, revealing actionable insights: wooden: 13/20 furniture-dominant; riding: 9/20 child-dominant; orange: 10/20 color-dominant. Visual grids in the additional files support model auditing and targeted fine-tuning, demonstrating downstream utility that CLIPScore and attribution maps cannot provide.
> See: https://github.com/anonymous1913/additional-files/raw/main/additional-file.pdf
>
> Table U4: Concept-level bias (K=20 per token).
>
> | Token   | Type      | Key insight                |
> |-|-|-|
> | dog     | noun      | Balanced, 8 contexts       |
> | orange  | adjective | 10/20 color-dominant       |
> | wooden  | adjective | 13/20 furniture-dominant   |
> | jumping | verb      | 11/20 person-dominant      |
> | riding  | verb      | 9/20 child-dominant        | |
>
> ---
> *W5/Q4*: We acknowledge the typos and will correct them in the revised manuscript.
>
> ---
> *W6/Q2*: Detailed experimental comparisons between our method and SAEs, conducted under the same settings, are provided in our response to Reviewer oY3J (W3).

---

> > ### Author Rebuttal · Reviewer_U2Tm · 2026-04-01
> >
> > The rebuttal has successfully addressed my concerns. I highly recommend incorporating these supplementary experiments and discussions into the main text and/or appendix of the revised manuscript, as they will be highly valuable for future readers. As promised, I am raising my score. Good luck.

---

> > > ### Author Response · Authors · 2026-04-05
> > >
> > > Dear Reviewer U2Tm,
> > >
> > > Thank you very much for your thoughtful comments, for carefully considering our rebuttal, and for updating your score. We are delighted that our responses have fully addressed your concerns and truly appreciate your recommendation to accept the paper.
> > >
> > > As suggested, we will incorporate all additional experiments and analyses discussed in the rebuttal into the supplementary material and final version to further strengthen the paper.
> > >
> > > Thank you once again for your positive feedback and support.

---

### Official Review · Reviewer_oY3J · 2026-03-07

**Soundness:** 3
**Presentation:** 3
**Significance:** 3
**Originality:** 3
**Overall Recommendation:** 4
**Confidence:** 4

**Summary:**

This paper proposes Neuro-FeX, a neuro-fuzzy framework for interpreting large multimodal models by turning hidden representations into human-readable concepts and fuzzy IF–THEN rules. It extracts token-specific internal features from models such as LLaVA and Qwen2-VL, decomposes them through fuzzification and defuzzification, and grounds the learned concepts in both images and text. The goal is not just to identify concepts, but to expose the latent reasoning structure behind model predictions. Experiments on COCO suggest improved multimodal grounding and, in some variants, better concept disentanglement, though the empirical evidence remains somewhat limited.

**Compliance With Llm Reviewing Policy:**

Affirmed.

**Final Justification:**

Thanks for the author's reply. My issues have been resolved, and I will raise my score.

**Key Questions For Authors:**

See above.

**Limitations:**

yes

**Strengths And Weaknesses:**

# Strengths
1. The paper goes beyond standard concept decomposition and tries to recover fuzzy IF–THEN rule structure from multimodal model representations, which is a more ambitious and interesting interpretability goal.
2. The method is clearly structured, with representation extraction, fuzzification, defuzzification, and multimodal grounding forming a fairly complete framework.
3. The learned concepts are grounded in both visual samples and decoded words, which makes the resulting explanations easy to inspect qualitatively.

# Weaknesses
1. The evidence does not fully support the main claim. The paper emphasizes rule-level interpretability and logical reasoning structure, but the evaluation mainly relies on CLIPScore, overlap, and qualitative examples, which are not strong enough to verify that the method truly captures the model’s internal reasoning process.
2. The empirical scope is too narrow. The experiments are conducted on COCO with only a small set of target tokens such as dog, cat, train, and bus, which makes the validation feel more like a case study than a broad demonstration of generality.
3. Model coverage is limited and somewhat dated. The paper validates mainly on LLaVA and Qwen2-VL-7B, which were reasonable choices, but not broad enough to support a strong general claim about modern LMM interpretability. A stronger paper would test on a wider range of newer multimodal models as well. Notably, Qwen2.5-VL was released in February 2025, and Qwen3-VL was released in September 2025, so there are already newer generations that are not covered here.

---

> ### Author Rebuttal · Authors · 2026-03-31
>
> We thank the reviewer for insightful questions, addressed as follows.
>
> W1: To substantiate that Neuro-FeX faithfully captures the model's internal reasoning, we validate the IF part (antecedents) and THEN part (consequents) independently.
>
> Experiment 1: Rule Consistency (IF part). If the Gaussian membership functions partition the representation space meaningfully, top-activating images per rule should form visually coherent groups. For each rule, we retrieve top-5 MAS and compute mean pairwise CLIP coherence, compared against 100 random-group baselines. Neuro-FeX achieves ConsDelta = +0.119 and 94% of rules (19/20) individually exceed the random baseline (Table T3), confirming that membership functions partition the space semantically, not arbitrarily.
>
> T3: Rule consistency (mean +/- std, 5 tokens). ConsDelta = CLIP coherence of top-5 MAS per rule minus random baseline.
>
> | Metric                              | Neuro-FeX        | Random baseline  |
> |-|-|-|
> | Mean CLIP coherence                 | 0.651 +/- 0.029  | 0.532 +/- 0.021  |
> | ConsDelta (coherence - random)      | +0.119           | -     |
> | Fraction of rules > random          | 94% (19/20)      | -      |
>
> Experiment 2: Faithfulness (THEN part). If firing strengths truely reflect the model's reasoning, top-$r$ rules should reconstruct the hidden state more faithfully than random-$r$ rules. To quantify it we follow standard faithfulness evaluation from the paper [2]. The faithfulness gap (top-$r$ minus random-$r$ LogitRatio) quantifies this. As reported in Table T4, top-$r$ rules carry $4.36\times$ more prediction signal at $r=1$ and $2.81\times$ at $r=5$, directly proving that firing strengths identify rules driving the model's output.
>
> T4: Faithfulness, LogitRatio (mean +/- std, 5 tokens). Faith gap = top-$r$ minus random-$r$ LogitRatio.
>
> | Condition | Neuro-FeX        | Random baseline  | Faith gap      |
> |-|-|-|-|
> | top-1     | 0.266 +/- 0.105  | 0.061 +/- 0.019  | +0.205 (4.36x) |
> | top-5     | 0.826 +/- 0.107  | 0.294 +/- 0.029  | +0.532 (2.81x) |
> | top-10    | 1.046 +/- 0.029  | 0.547 +/- 0.061  | +0.499 (1.91x) |
>
> ---
>
> W2 & W3:
> To expand our empirical scope, we tested 30 diverse COCO tokens (see Q2, reviewer-Ojzs), computed MS scores, and compared Neuro-FeX to SAE baselines across various models using Imagenet. Results below:
>
> We conducted a direct comparison with BatchTopK SAE and Matryoshka SAE under identical settings: same training data (full ImageNet, no token filtering), same vision encoders, same MS formulation, and identical dictionary size $K=1024$ [1]. Although Neuro-FeX activations ($V_{:,k}$, shaped by RBF membership) and SAE activations (TopK encoder outputs) differ architecturally, both quantify how strongly each input activates a concept, and MS is computed identically for all methods. MS is validated at 82.8% human alignment (1,000 questions, 71 annotators). We followed similar experimental setup details as [1].
>
> Neuro-FeX achieves the highest Max MS across all three encoders and the highest Mean MS on CLIP ViT-L and SigLIP SoViT-400m (Table T5). On SigLIP, mean MS = 0.711 vs 0.610 for both SAE baselines (+16.5%). On CLIP ViT-L, mean MS = 0.188 vs Matryoshka SAE 0.160 (+17.2%). On Qwen3, mean MS (0.544) is within standard deviation of Matryoshka SAE (0.549, diff = 0.005), while achieving superior Max MS (0.918 vs 0.889). Neuro-FeX consistently outperforms BatchTopK SAE and CoX-LMM on all backbones. Since $K=1024$ for all methods, the only variable is the decomposition method itself. The improvement reflects a principled architectural advantage of the proposed Neuro-FeX: RBF membership functions enforce geometric locality, ensuring concept $k$ activates only for images near center $c_k$, embedding monosemanticity directly into the activation mechanism.
>
> Table T5: MonoSemanticity Score (MS, range [0,1], K=1024 for all methods).
>
> | Model               | Method          | Mean MS +/- std | Max MS |
> |-|-|-|-|
> | CLIP ViT-L          | Neuro-FeX       | 0.188 +/- 0.18  | 0.860  |
> |                     | Matryoshka SAE  | 0.160 +/- 0.17  | 0.820  |
> |                     | CoX-LMM         | 0.102 +/- 0.05  | 0.695  |
> |                     | BatchTopK SAE   | 0.120 +/- 0.11  | 0.570  |
> | SigLIP SoViT-400m   | Neuro-FeX       | 0.711 +/- 0.06  | 0.923  |
> |                     | Matryoshka SAE  | 0.610 +/- 0.09  | 0.899  |
> |                     | BatchTopK SAE   | 0.610 +/- 0.09  | 0.879  |
> |                     | CoX-LMM         | 0.589 +/- 0.05  | 0.744  |
> | Qwen3               | Neuro-FeX       | 0.544 +/- 0.06  | 0.918  |
> |                     | Matryoshka SAE  | 0.549 +/- 0.10  | 0.889  |
> |                     | BatchTopK SAE   | 0.528 +/- 0.10  | 0.845  |
> |                     | CoX-LMM         | 0.473 +/- 0.06  | 0.733  |
>
>
> ---
>
> [1] Pach M, et al. Sparse autoencoders learn monosemantic features in vision-language models[J]. NeurIPS 2025.
> [2] W. Samek et al., "Explaining deep neural networks and beyond," Proc. IEEE, 109(3), 2021.

---

> > ### Author Rebuttal · Reviewer_oY3J · 2026-04-03
> >
> > Thanks for the author's reply. My issues have been resolved, and I will raise my score.

---

> > > ### Author Response · Authors · 2026-04-05
> > >
> > > Dear Reviewer oY3J,
> > >
> > > Thank you very much for your thoughtful comments, for carefully considering our rebuttal, and for updating your score. We are delighted that our responses have fully addressed your concerns and truly appreciate your recommendation to accept the paper.
> > >
> > > Thank you once again for your positive feedback and support.

---

### Official Review · Reviewer_Qjzs · 2026-03-13

**Soundness:** 3
**Presentation:** 3
**Significance:** 3
**Originality:** 2
**Overall Recommendation:** 4
**Confidence:** 3

**Summary:**

The paper introduces a novel interpretability framework for Large Multimodal Models utilizing a Human-Inspired (Neuro-fuzzy) approach for learning multimodal token representations. The proposed mechanism allows to derive “multimodal concepts” that are both semantically coherent and interpretable. The authors evaluate their approach using Clip-score similarity showing these concepts are quite robust across test samples, and semantically-rich.

**Compliance With Llm Reviewing Policy:**

Affirmed.

**Final Justification:**

I would like to increase my score after rebuttal. In the begining, I noted that the original version of the paper is interesting, but misses the important comparison with the strongest baselines in this domain. During the rebuttal, the authors covered most of the concerns raised by me and other reviewers, thus, the paper now sounds solid, and I suppose that it can be considered for accepting on the conference.

**Key Questions For Authors:**

1) The method is formulated for a target token t, and the authors explicitly build a filtered dataset of images where t appears in both the prediction and the ground-truth caption, then decompose the resulting token-specific representation matrix Z. Does this mean that Neuro-FeX must be refit separately for every new target token, or is there any shared concept space that transfers across tokens?
2) How confident should we be that the method generalizes beyond these four concrete object tokens?
3) Why is there no comparison with sparse autoencoders?

**Limitations:**

The main limitations are the following:
1) Limited target semantic classes evaluation. It is not evident for now, how robust the introduced approach is.
2) Limited comparison with the top-level methods of interpretability, such as SAE.

While, basically, I like the paper. If the authors address my concerns I am willing to increase the score.

**Strengths And Weaknesses:**

1) The paper is reasonably technically sound. It proposes a token-conditioned neuro-fuzzy decomposition of LMM hidden representations into a concept dictionary and normalized firing strengths, and the overall pipeline is described clearly enough to understand what is being done. The experiments are quite extensive, however, in the weaknesses part I mention some points that are underevaluated in my opinion.
2) The paper is overall easy to follow. The motivation is clear, and figures and tables support the claim.
3) The paper does not introduce a completely new decomposition family, but the combination of concept extraction with a neuro-fuzzy, TSK-style reasoning view is interesting and reasonably original. In particular, grounding learned concepts in both visual samples and decoded text, while also interpreting activations as fuzzy-rule firing strengths can be used as a novel interpretation framework for multimodal models.

The main weaknesses of the paper, as I see them, are the following:
1) In practice, the paper evaluates only four target tokens: dog, cat, train, and bus, and the whole procedure is built in a token-specific way, since the authors first collect samples where a chosen target token appears in both prediction and ground truth and then decompose the corresponding representation matrix. Because of this, it is not clear whether the method will generalize to a broader vocabulary, especially to rarer, more abstract, or relational tokens. As far as I understand, for a new token one would likely need to rerun the explainer pipeline.
2) Limited evaluation with baselines. The paper compares against CoX-LMM and also reports overlap relative to PCA, K-means, and Semi-NMF, but there is no comparison with sparse autoencoders. Since the current method is already quite close in spirit to sparse dictionary learning, an SAE (as a top-performing concept revealing tool) baseline would be a valuable comparison.

Note, there is a typo in abstract "Huamn-Inspired" -> "Human-Inspired"

---

> ### Author Rebuttal · Authors · 2026-03-31
>
> We thank the reviewer for their insightful questions, addressed below (W: Weakness, Q: Question).
>
> W1/Q1: Our setup is intentionally token-specific to isolate the visual concepts driving the prediction of a target token $t$. However, Neuro-FeX is not restricted to token-level operations, as demonstrated by two complementary experiments.
>
> Exp1: We apply Neuro-FeX to five tokens across three linguistic categories (noun: dog; adjectives: orange, wooden; verbs: jumping, riding), learning $K=20$ concept vectors per token via fuzzy decomposition of token-specific hidden states. Because a single LMM processes all tokens, their concept dictionaries $U_t$ share an ambient representation space. We identify shared concepts via Hungarian matching between dictionaries $U_t$ and $U_{t'}$, finding optimal 1-to-1 pairings based on cosine similarity (Table T1).
>
> Same-type token pairs (verb$\leftrightarrow$verb: 0.352,
> adjective$\leftrightarrow$adjective: 0.502) share significantly more concepts than
> cross-type pairs (mean: 0.271), demonstrating that the LMM organises its
> representations along linguistic category lines, a structure that Neuro-FeX implicitly
> reveals through independent per-token decompositions. Qualitative inspection of
> grounded words $T_j$ confirms that shared concept groups are semantically
> interpretable. For example: the token *jumping* (fuzzy rule 6) and *riding* (fuzzy rule 18), both action verbs,
> ground in dynamic motion words (*catching, skating, balancing*), capturing the
> same visual concept of dynamic human motion despite being learned from entirely
> separate token-specific datasets. Visual concept grids, corresponding to these tokens are provided
> in the additional files (Link: https://github.com/anonymous1913/additional-files/raw/main/additional-file.pdf).
>
> T1: This table reports the mean Hungarian matching between two concept dictionaries $U_t$ and $U_{t'}$.
>
> |         | dog   | orange | wooden | jumping | riding |
> |-|-|-|-|-|-|
> | dog     | 1.000 | 0.454  | 0.465  | 0.176   | 0.253  |
> | orange  | 0.454 | 1.000  | 0.502  | 0.171   | 0.220  |
> | wooden  | 0.465 | 0.502  | 1.000  | 0.158   | 0.231  |
> | jumping | 0.176 | 0.171  | 0.158  | 1.000   | 0.352  |
> | riding  | 0.253 | 0.220  | 0.231  | 0.352   | 1.000  |
>
> Exp 2 (Token-independent): To demonstrate that Neuro-FeX generalizes beyond token-specific settings, we apply it directly to the full ImageNet dataset without token filtering, jointly decomposing all hidden states. We evaluate using the MonoSemanticity (MS) score [1], measuring the visual coherence of top-activating images, which aligns with human perception at 82.8% (1,000 queries, 71 annotators). As shown in Table T5 (Reviewer oY3J), Neuro-FeX outperforms both SAE baselines in MS across three vision encoders, confirming it learns highly selective, monosemantic concepts globally. This establishes that Neuro-FeX's fuzzy membership structure, driven by RBF geometric locality, produces more focused and coherent activations than SAE sparsity alone, whether applied to specific tokens or the full vocabulary.
>
> ---
>
> *Q2/W1*: We thank the reviewer for raising this important point. To validate that our method generalizes beyond the initial four concrete object tokens, we have conducted additional experiments on a much larger and more diverse set of 30 tokens. To ensure comprehensive linguistic coverage, this expanded set includes nouns, verbs, adjectives, and abstract concepts (e.g., wooden, orange, bird, cow, umbrella, backpack, sitting, eating, flying, standing, dark, bright, small, inside, together, behind, holding).
>
> We evaluated this diverse set of tokens using the Qwen2.5 and Qwen3 models. In Table T2, we provide the mean CLIP scores comparing the baseline CoXLMM with our proposed model, Neuro-FeX. As the results clearly demonstrate, Neuro-FeX maintains strong performance across these varied linguistic structures, confirming that our framework successfully generalizes far beyond the initial four simple tokens.
>
> T2: Mean CLIPScore across 30 diverse tokens comparing Neuro-FeX variants against the CoX-LMM baseline.
>
> | Model       | Neuro-FeX-v0 | Neuro-FeX-v1 | Neuro-FeX-v2 | CoX-LMM |
> |-|-|-|-|-|
> | Qwen-2.5-VL | 0.621        | **0.643**        | 0.624        | 0.616   |
> | Qwen-3-VL   | 0.590        | **0.592**       | 0.587        | 0.582   |
>
> ---
>
> *Q3/W2*: Detailed experimental comparisons between our method and SAEs [1], conducted under the same settings, are provided in our response to **W3** of the reviewer oY3J, due to character limit in responses.
>
> ---
>
> References:
> [1] Pach M, Karthik S, Bouniot Q, et al. Sparse autoencoders learn monosemantic features in vision-language models[J]. NeurIPS 2025.

---

> > ### Author Rebuttal · Reviewer_Qjzs · 2026-04-05
> >
> > I thank the authors for the response, they cover all the issues raised. I am willing to increase my score.

---

> > > ### Author Response · Authors · 2026-04-05
> > >
> > > Dear Reviewer Qjzs,
> > >
> > > Thank you very much for your thoughtful comments, for carefully considering our rebuttal, and for updating your score. We are delighted that our responses have fully addressed your concerns and truly appreciate your recommendation to accept the paper.
> > >
> > > Thank you once again for your positive feedback and support.

---

### Decision · Program_Chairs · 2026-04-30

**Decision:**

Accept (regular)

**Comment:**

This paper introduces a framework utilizing a Human-Inspired (Neuro-fuzzy) approach for learning token representations, making the interpretation of learned representations directly through explicit logic. The method is validated through qualitative and quantitative experiments, demonstrating the utility of these concepts in interpreting test samples. All reviewers find the paper interesting and the topic important. There are some initial concerns on the limited evaluations and scope of the experiments and the method, as well as some technical details. The rebuttal successfully addresses these concerns. All reviewers are convinced and raise/maintain their positive score. I recommend accept.